# Unsupervised Learning for Solving the Travelling Salesman Problem

**Yimeng Min**[*]
Dept. of Computer Science
Cornell University
Ithaca, NY, USA
min@cs.cornell.edu

**Yiwei Bai**[*]
Dept. of Computer Science
Cornell University
Ithaca, NY, USA
bywbilly@gmail.com

**Carla P. Gomes**
Dept. of Computer Science
Cornell University
Ithaca, NY, USA
gomes@cs.cornell.edu

## Abstract

We propose UTSP, an Unsupervised Learning (UL) framework for solving the Travelling Salesman Problem (TSP). We train a Graph Neural Network (GNN) using a surrogate loss. The GNN outputs a heat map representing the probability for each edge to be part of the optimal path. We then apply local search to generate our final prediction based on the heat map. Our loss function consists of two parts: one pushes the model to find the shortest path and the other serves as a surrogate for the constraint that the route should form a Hamiltonian Cycle. Experimental results show that UTSP outperforms the existing data-driven TSP heuristics. Our approach is parameter efficient as well as data efficient: the model takes $\sim$ 10% of the number of parameters and $\sim$ 0.2% of training samples compared with Reinforcement Learning or Supervised Learning methods.

## 1   Introduction

Euclidean Travelling Salesman Problem (TSP) is one of the most famous and intensely studied NP-hard problems in the combinatorial optimization community. Exact methods, such as Concorde Applegate et al. [2006], use the cutting-plane method, iteratively solving linear programming relaxations of the TSP. These methods are usually implemented within a branch-and-cut framework, integrating the cutting-plane algorithm into a branch-and-bound search. While these exact methods can find solutions with guaranteed optimality for up to tens of thousands of nodes, the execution time can be exceedingly expensive. A different strategy is using heuristics such as LKH Helsgaun [2000]. These heuristics aim to find near-optimal solutions with a notable reduction in time complexity. Typically, they are manually crafted, drawing upon expert insights and domain-specific knowledge.

Recently, the success of Graph Neural Networks (GNNs) for a variety of machine learning tasks has sparked interests in building data-driven heuristics for approximating TSP solutions. For example, Kwon et al. [2020] uses a data-driven approach known as Policy Optimization with Multiple Optima (POMO), POMO relies on Reinforcement Learning (RL) and avoids the utilization of hand-crafted heuristics. Qiu et al. [2022] proposes a Meta-Learning framework which enhances the stability of RL training. Sun and Yang [2023] applies Supervised Learning (SL) and adopts a graph-based diffusion framework. Additionally, the authors use a cosine inference schedule to improve the efficiency of their model. Joshi et al. [2019a] trains their GNN model in a SL fashion, and the model outputs an edge adjacency matrix that indicates the likelihood of edges being part of the TSP tour. The edge predictions form a heat map, which is transformed into a valid tour through a beam search method. Similarly, Fu et al. [2021] trains the GNN model on small sub-graphs to generate the corresponding heat maps using SL. These small heat maps are then integrated to build a large final heat map.

---

[*]Equal contribution

37th Conference on Neural Information Processing Systems (NeurIPS 2023).

Overall, these learning-based models usually build heuristics by reducing the length of TSP tours via RL or directly learning from the optimal solutions via SL. However, since TSP is NP-hard, SL can cause expensive annotation problems due to the costly search time involved in generating optimal solutions. For RL, when dealing with big graphs, the model will run into the sparse reward problem because the reward is decided after decoding a complete solution. The sparse reward problem results in poor generalization performance and high training variance. Furthermore, both RL and SL suffer from expensive large-scale training. These models take more than one million training samples when dealing with TSP with 100 nodes, making the training process very time-consuming.

## 2 Our Model

In this work, we build a data-driven TSP heuristic in an Unsupervised Learning (UL) fashion and generate the heat map non-autoregressively. We construct a surrogate loss function with two parts: one encourages the GNN to find the shortest path, and the other acts as a proxy for the constraint that the path should be a Hamiltonian Cycle over all nodes. The surrogate loss enables us to update the model without decoding a complete solution. This helps alleviate the sparse reward problem encountered in RL, and thus, it avoids unstable training or slow convergence Kool et al. [2019]. Our UTSP method does not rely on labelled data, which helps the model avoid the expensive annotation problems encountered in SL and significantly reduces the time cost. In fact, due to the prohibitive time cost of building training datasets for large instances, many SL methods are trained on relatively small instances only Fu et al. [2021]Joshi et al. [2019a]. Such SL models scale poorly to big instances, while with our UTSP model, we can train our model on larger instances directly. Overall, our training does not rely on any labelled training data and converges faster compared to RL/SL methods.

The model takes the coordinates as the input of GNN. The distance between two nodes determines the edge weight in the adjacency matrix. After training the GNN, the heat map is converted to a valid tour using local search. We evaluate the performance of UTSP through comparisons on TSP cases of fixed graph sizes up to 1,000 nodes. We note that UTSP is fundamentally different from RL, which may also be considered unsupervised. While RL requires a Markov Decision Process (MDP), and its reward is extracted after obtaining solutions, our method does not use a MDP and the loss function (reward) is determined based on a heat map.

Overall, UTSP requires only a small amount of (unlabelled) data and compensates for it by employing an unsupervised surrogate loss function and an expressive GNN. The heat maps built using UTSP help reduce the search space and facilitate the local search. We further show that the expressive power of GNNs is critical for generating non-smooth heat maps.

## 3 Methodologies

In this paper, we study symmetric TSP on a 2D plane. Given $n$ cities and the coordinates $(x_i, y_i) \in \mathbb{R}^2$ of these cities, our goal is to find the shortest possible route that visits each city exactly once and returns to the origin city, where $i \in \{1, 2, 3, ..., n\}$ is the index of the city.

### 3.1 Graph Neural Network

Given a TSP instance, let $\mathbf{D}_{i,j}$ denote the Euclidean distance between city $i$ and city $j$. $\mathbf{D} \in \mathbb{R}^{n \times n}$ is the distance matrix. We first build adjacency matrix $\mathbf{W} \in \mathbb{R}^{n \times n}$ with $\mathbf{W}_{i,j} = e^{-\mathbf{D}_{i,j}/\tau}$ and node feature $\mathbf{F} \in \mathbb{R}^{n \times 2}$ based on the input coordinates, where $\mathbf{F}_i = (x_i, y_i)$ and $\tau$ is the temperature. The node feature matrix $\mathbf{F}$ and the weight matrix $\mathbf{W}$ are then fed into a GNN to generate a soft indicator matrix $\mathbb{T} \in \mathbb{R}^{n \times n}$.

In our model, we use Scattering Attention GNN (SAG), SAG has both low-pass and band-pass filters and can build adaptive representations by learning node-wise weights for combining multiple different channels in the network using attention-based architecture. Recent studies show that SAG can output expressive representations for graph combinatorial problems such as maximum clique and remain lightweight Min et al. [2022].

Let $\mathcal{S} \in \mathbb{R}^{n \times n}$ denote the output of SAG, we first apply a column-wise Softmax activation to the GNN's output and we can summarize this operation in matrix notation as $\mathbb{T}_{i,j} = e^{\mathcal{S}_{i,j}} / \sum_{k=1}^{n} e^{\mathcal{S}_{k,j}}$.

This ensures that each element in $\mathbb{T}$ is greater than zero and the summation of each column is 1. We then use $\mathbb{T}$ to build a heat map $\mathcal{H}$, where $\mathcal{H} \in \mathbb{R}^{n \times n}$.

In our model, we use $\mathcal{H}$ to estimate the probability of each edge belonging to the optimal solution and use $\mathbb{T}$ to build a surrogate loss of the Hamiltonian Cycle constraint. This will allow us to build a non-smooth heat map $\mathcal{H}$ and improve the performance of the local search.

## 3.2 Building the Heat Map using the soft indicator matrix

Before building the unsupervised loss, let's recall the definition of TSP. The objective of TSP is to identify the shortest Hamiltonian Cycle of a graph. Therefore, the unsupervised surrogate loss should act as a proxy for two requirements: the Hamiltonian Cycle constraint and the shortest path constraint. However, designing a surrogate loss for the Hamiltonian Cycle constraint can be challenging, particularly when working with a heat map $\mathcal{H}$. To address this, we introduce the $\mathbb{T} \to \mathcal{H}$ transformation, which enables the model to implicitly encode the Hamiltonian Cycle constraint.

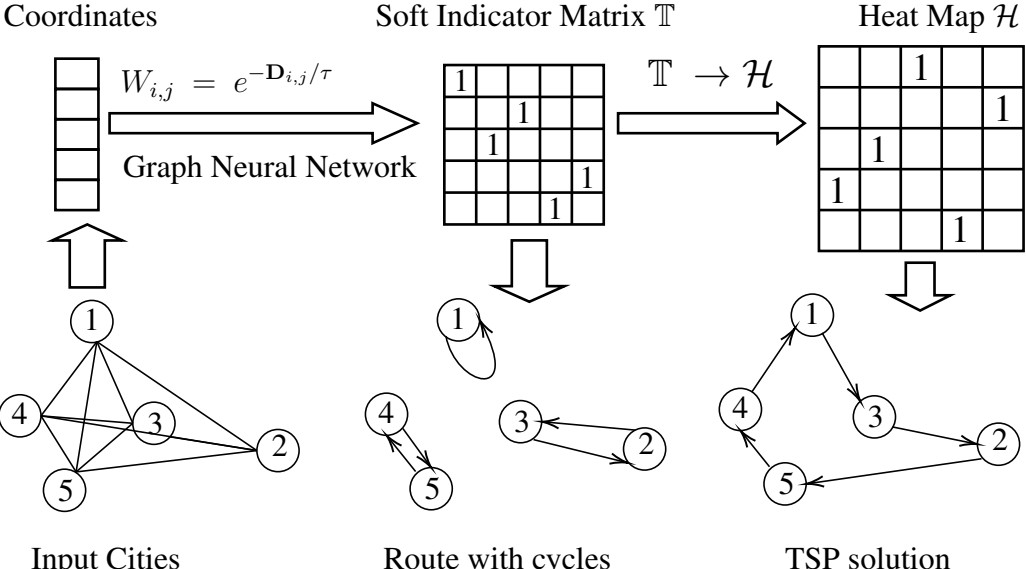

Figure 1: We use a SAG to generate a non-smooth soft indicator matrix $\mathbb{T}$. The SAG model is a function of coordinates and weighted adjacency matrix. We then build the heat map $\mathcal{H}$ based on $\mathbb{T}$ using the transformation in Equation 1.

To better understand the $\mathbb{T} \to \mathcal{H}$ transformation, we show a binary instance in Figure 1. Figure 1 illustrates a soft indicator matrix $\mathbb{T}$, its heat map $\mathcal{H}$ following the transformation $\mathbb{T} \to \mathcal{H}$, and their corresponding routes. When we directly use the soft indicator matrix $\mathbb{T}$ as the heat map. It can result in loops (parallel edges) between cities, such as (2,3) and (4,5) in Figure 1 (middle). After we apply the $\mathbb{T} \to \mathcal{H}$ transformation, the corresponding heat $\mathcal{H}$ is a Hamiltonian Cycle, as shown in the right part in Figure 1.

## 3.3 $\mathbb{T} \to \mathcal{H}$ transformation

We build the heat map $\mathcal{H}$ based on $\mathbb{T}$. As mentioned, $\mathcal{H}_{i,j}$ is the probability for edge $(i,j)$ to belong to the optimal TSP solution. We define $\mathcal{H}$ as:

$$\mathcal{H} = \sum_{t=1}^{n-1} p_t p_{t+1}^T + p_n p_1^T, \tag{1}$$

where $p_t \in \mathbb{R}^{n \times 1}$ is the $t_{th}$ column of $\mathbb{T}$, $\mathbb{T} = [p_1|p_2|...|p_n]$. As shown in Figure 1, the first row in $\mathcal{H}$ is the probability distribution of directed edges start from city 1, and since the third element is the only non-zero one in the first row, we then add directed edge $1 \rightarrow 3$ to our TSP solution. Similarly, the first column in $\mathcal{H}$ can be regarded as the probability distribution of directed edges which end in city 1. Ideally, given a graph $\mathcal{G}$ with $n$ nodes, we want to build a soft indicator matrix where each row and column are assigned with one value 1 (True) and $n - 1$ values 0 (False), so that the heat map will only contain one valid solution. In practice, we will build a soft indicator matrix $\mathbb{T}$ whose heat map $\mathcal{H}$ assigns large probabilities to the edges in the TSP solution and small probabilities to the other edges.

Overall, the $\mathbb{T} \rightarrow \mathcal{H}$ transformation in Equation 1 enables us to build a proxy for the Hamiltonian Cycle constraint. We further prove that $\mathcal{H}$ represents one Hamiltonian Cycle when each row and column in $\mathbb{T}$ have one value 1 (True) and $n - 1$ value 0 (False). We refer to the proof in appendix D.

### 3.4 Unsupervised Loss

In order to generate such an expressive soft indicator matrix $\mathbb{T}$, we minimize the following objective function:

$$\mathcal{L} = \lambda_1 \underbrace{\sum_{i=1}^{n} (\sum_{j=1}^{n} \mathbb{T}_{i,j} - 1)^2}_{\text{Row-wise constraint}} + \lambda_2 \underbrace{\sum_{i}^{n} \mathcal{H}_{i,i}}_{\text{No self-loops}} + \underbrace{\sum_{i=1}^{n} \sum_{j=1}^{n} \mathbf{D}_{i,j} \mathcal{H}_{i,j}}_{\text{Minimize the distance}} . \tag{2}$$

The first term in $\mathcal{L}$ encourages the summation of each row in $\mathbb{T}$ to be close to 1. As mentioned, we normalize each column of $\mathbb{T}$ using Softmax activation. So when the first term is minimized to zero, each row and column in $\mathbb{T}$ are normalized (doubly stochastic). The second term penalizes the weight on the main diagonal of $\mathcal{H}$, this discourages self-loops in TSP solutions. The third term can be regarded as the expected TSP length of the heat map $\mathcal{H}$, where $\mathbf{D}_{i,j}$ is the distance between city $i$ and $j$. As mentioned, since $\mathcal{H}$ corresponds to a Hamiltonian Cycle given an ideal soft indicator matrix with one value 1 (True) and $n - 1$ value 0 (False) in each row and column. Then the minimum value of $\sum_{i=1}^{n} \sum_{j=1}^{n} \mathbf{D}_{i,j} \mathcal{H}_{i,j}$ is the shortest Hamiltonian Cycle on the graph, which is the optimal solution of TSP. To summarize, we build a loss function which contains both the *shortest* and the *Hamiltonian Cycle* constraints. The *shortest* constraint is realized by minimizing $\sum_{i=1}^{n} \sum_{j=1}^{n} \mathbf{D}_{i,j} \mathcal{H}_{i,j}$. For the *Hamiltonian Cycle* constraint, instead of writing it in a Lagrangian relaxation style penalty, we use a GNN which encourages a non-smooth representation, along with the doubly stochastic penalty, and the $\mathbb{T} \rightarrow \mathcal{H}$ transformation.

### 3.5 Edge Elimination

Given a heat map $\mathcal{H}$, we consider $M$ largest elements in each row (without diagonal elements) and set other $n - M$ elements as 0. Let $\tilde{H}$ denote the new heat map, we then symmetrize the new heat map by $\mathcal{H}' = \tilde{H} + \tilde{H}^T$. Let $\mathbf{E}_{ij} \in \{0, 1\}$ denote whether an undirected edge $(i, j)$ is in our prediction or not. Without loss of generality, we can assume $0 < i < j \leq n$ and define $\mathbf{E}_{ij}$ as :

$$\mathbf{E}_{ij} = \begin{cases} 1, & \text{if } \mathcal{H}'_{ij} = \mathcal{H}'_{ji} > 0 \\ 0, & \text{otherwise} \end{cases} .$$

Let $\Pi$ denote the set of undirected edges $(i, j)$ with $\mathbf{E}_{ij} = 1$. Ideally, we would build a prediction edge set $\Pi$ with a small $M$ value, and $\Pi$ can cover all the ground truth edges so that we can reduce search space size from $n(n - 1)/2$ to $|\Pi|$. In practice, we aim to let $\Pi$ cover as many ground truth edges as possible and use $\mathcal{H}'$ to guide the local search process.

## 4 Local Search

### 4.1 Heat Map Guided Best-first Local Search

We employ the best-first local search guided by the heat map to generate the final solution. Best-first search is a heuristic search that explores the search space by expanding the most promising node selected w.r.t. an evaluation function $f(node)$. In our framework, each node of the search tree is a

complete TSP solution. For the initialization of one search tree, we randomly generate a valid TSP solution and improve it using the 2-opt heuristic until no better solution is found. The expand action of the search node refers to Fu et al. [2021] and is based on the widely used k-opt heuristic Croes [1958], where it replaces $k$ old edges (in the current solution) with $k$ new edges, i.e., transforms the old solution to a new solution. More formally, we use a series of cities, $u_1, v_1, u_2, \ldots, u_k, v_{k+1}$, to represent an action, where $v_{k+1} = u_1$ to ensure it is a valid solution. All the edges $(u_i, v_i)$ $(1 \leq i \leq k)$ are removed from the tour and $(v_i, u_{i+1})(1 \leq i \leq k)$ are added to the tour. Note that once we know $u_i$, $v_i$ is deterministically decided. $u_1$ is randomly selected for each expansion, and $v_1$ is decided subsequently. Then we select $u_{i+1}(i \geq 1)$ as follows: (1) if $u_{i+1} = u_1$, i.e., forms a new TSP tour, leads to an improved solution then we set $u_{i+1} = u_1$ and have a candidate solution. (2) if $i \geq K$, then we will discard this action and start a new expand action, where $K$ is a hyper-parameter which controls the maximal edges we can remove in one action. (3) otherwise, we select $u_{i+1}$ based on the heat map stochastically. We use $N_{u,v}$ to denote the times the edge $(u,v)$ is selected during the entire search procedure. The likelihood of selecting the edge $(u,v)$ is denoted by $L_{u,v} = \mathcal{H}'_{u,v} + \alpha\sqrt{\frac{\log(S+1)}{N_{u,v}+1}}$, where $\alpha$ is a hyper-parameter and $S$ is the local search's total number of expand actions. The first term encourages the algorithm to select the edge with a high heat map value, while the second term diversifies the selected edges. Moreover, when selecting the city $v$ given $u$, we only consider the cities from the candidate set of $v$. This candidate set consists of cities with the top $M$ heat map value or the nearest $M$ cities

Among all the possible new solutions, we use the tour's length as the evaluation function $f$, i.e., we select the solution of the shortest tour length as the next search node. For each search node, we try at most $T$ expand actions. From these $T$ expand actions, if no improved solution is found, we randomly generate a new initial solution and start another round of best-first local search.

### 4.2 Updating the Heat Map

We borrow the idea of the backpropagation used in Monte Carlo Tree Search (MCTS). We use $s$ to denote the current search node, $s'$ to denote the next search node ($s'$ has to improve $s$), and $L(s)$ to represent the tour length of node $s$. The heat map $\mathcal{H}'$ is updated as:

$$\mathcal{H}'_{v_i, u_{i+1}} = \mathcal{H}'_{v_i, u_{i+1}} + \beta[\exp(\frac{L(s) - L(s')}{L(s)}) - 1],$$

where $\beta$ is a search parameters and $(v_i, u_{i+1})(1 \leq i \leq k)$ is the actions used to transform $s$ to $s'$. We raise the importance of the edges that lead to a better solution. If we cannot find an improved solution for the current node, then no update is executed for the heat map.

### 4.3 Leveraging Randomness

Randomness is shown to be very powerful in the local search community Gomes et al. [1998], Bresina [1996], Gomes et al. [2000]. Our local search procedure also employs it to improve performance. When we stop the current round of local search by not finding an improved solution within $T$ expand actions and switch to a new best-first local search with a new initial solution, we randomly modify the parameter $K$. A larger $K$ value results in more time searching from one initial solution. The intuition is that sometimes we want more initial solutions while sometimes we want to search deeper (replace more edges in k-opt) for a specific solution. Besides that, we also randomly decide how we construct the candidate set (based on the heat map or pairwise distance) for each city before the new round of local search.

## 5 Experiments

### 5.1 Dataset

Our dataset contains 2,000 samples for training and 1,000 samples for validation. We use the same test dataset in Fu et al. [2021]. The test dataset contains $10,000$ 2D-Euclidean TSP instances for $n = 20, 50, 100$, and 128 instances for $n = 200, 500, 1,000$. We train our models on TSP instances with 20, 50, 100, 200, 500, and 1,000 vertices. We then build the corresponding heat maps based on these trained models.

Table 1: Results of SAG + Local Search w.r.t. existing baselines. We evaluate the models on 10,000 TSP 20 instances and 10,000 TSP 50 instances. Baselines include GAT-RL1: Deudon et al. [2018]; GAT-RL2: Kool et al. [2019]; GCN-SL1: Joshi et al. [2019a]; POMO: Kwon et al. [2020] and Att-GCRN Fu et al. [2021].

| Method | Type | TSP20 | | | TSP50 | | |
| --- | --- | --- | --- | --- | --- | --- | --- |
| | | Length | Gap (%) | Time | Length | Gap (%) | Time |
| Concorde | Solver | 3.8303 | 0.0000 | 2.31m | 5.6906 | 0.0000 | 13.68m |
| Gurobi | Solver | 3.8302 | -0.0001 | 2.33m | 5.6905 | 0.0000 | 26.20m |
| LKH3 | Heuristic | 3.8303 | 0.0000 | 20.96m | 5.6906 | 0.0008 | 26.65m |
| GAT-RL1 | RL, S | 3.8741 | 1.1443 | 10.30m | 6.1085 | 7.3438 | 19.52m |
| GAT-RL1 | RL, S 2-OPT | 3.8501 | 0.5178 | 15.62m | 5.8941 | 3.5759 | 27.81m |
| GAT-RL2 | RL, S | 3.8322 | 0.0501 | 16.47m | 5.7185 | 0.4912 | 22.85m |
| GAT-RL2 | RL, G | 3.8413 | 0.2867 | 6.03s | 5.7849 | 1.6568 | 34.92s |
| GAT-RL2 | RL, BS | 3.8304 | 0.0022 | 15.01m | 5.7070 | 0.2892 | 25.58m |
| GCN-SL1 | SL, G | 3.8552 | 0.6509 | 19.41s | 5.8932 | 3.5608 | 2.00m |
| GCN-SL1 | SL, BS | 3.8347 | 0.1158 | 21.35m | 5.7071 | 0.2905 | 35.13m |
| GCN-SL1 | SL, BS* | 3.8305 | 0.0075 | 22.18m | 5.6920 | 0.0251 | 37.56m |
| POMO | RL | 3.83 | 0.00 | 3s | 5.69 | 0.03 | 16s |
| Att-GCRN | SL+RL MCTS | 3.8300 | **-0.0074** | 23.33s + 1.05m | 5.6908 | 0.0032 | 2.59m + 2.63m |
| **UTSP (ours)** | UL, Search | 3.8303 | -0.0009 | 38.23s + 1.04m | 5.6894 | **-0.0200** | 1.34m+ 2.60m |

## 5.2 Results

Table 2: Results of SAG + Local Search w.r.t. existing baselines. We evaluate the models on 10,000 TSP 100 instances and 128 TSP 200 instances. Baselines include GAT-RL1: Deudon et al. [2018]; GAT-RL2: Kool et al. [2019]; GCN-SL1: Joshi et al. [2019a]; POMO: Kwon et al. [2020] and Att-GCRN Fu et al. [2021].

| Method | Type | TSP100 | | | TSP200 | | |
| --- | --- | --- | --- | --- | --- | --- | --- |
| | | Length | Gap (%) | Time | Length | Gap (%) | Time |
| Concorde | Solver | 7.7609 | 0.0000 | 1.04h | 10.7191 | 0.0000 | 3.44m |
| Gurobi | Solver | 7.7609 | 0.0000 | 3.57h | 10.7036 | -0.1446 | 40.49m |
| LKH3 | Heuristic | 7.7611 | 0.0026 | 49.96m | 10.7195 | 0.0040 | 2.01m |
| GAT-RL1 | RL, S | 8.8372 | 13.8679 | 47.78m | 13.1746 | 22.9079 | 4.84m |
| GAT-RL1 | RL, S 2-OPT | 8.2449 | 6.2365 | 4.95h | 11.6104 | 8.3159 | 9.59m |
| GAT-RL2 | RL, S | 7.9735 | 2.7391 | 1.23h | 11.4497 | 6.8160 | 4.49m |
| GAT-RL2 | RL, G | 8.1008 | 4.3791 | 1.83m | 11.6096 | 8.3081 | 5.03s |
| GAT-RL2 | RL, BS | 7.9536 | 2.4829 | 1.68h | 11.3769 | 6.1364 | 5.77m |
| GCN-SL1 | SL, G | 8.4128 | 8.3995 | 11.08m | 17.0141 | 58.7272 | 59.11s |
| GCN-SL1 | SL, BS | 7.8763 | 1.4828 | 31.80m | 16.1878 | 51.0185 | 4.63m |
| GCN-SL1 | SL, BS* | 7.8719 | 1.4299 | 1.20h | 16.2081 | 51.2079 | 3.97m |
| POMO | RL | 7.77 | 0.14 | 1m | - | - | - |
| Att-GCRN | SL+RL MCTS | 7.7616 | 0.0096 | 3.94m + 5.25m | 10.7358 | 0.1563 | 20.62s + 1.33m |
| **UTSP (ours)** | UL, Search | 7.7608 | **-0.0011** | 5.68m + 5.21m | 10.7289 | **0.0918** | 0.56m+ 1.11m |

Table 1, Table 2 and Table 3 present model's performance on TSP 20, 50, 100, 200, 500 and 1,000. The first three lines in the tables summarize the performance of two exact solvers (Concorde and Gurobi) and LKH3 heuristic Helsgaun [2017]. The learning-based methods can be divided into RL sub-category and SL sub-category. Greedy decoding (G), Sampling (S), Beam Search (BS), and Monte Carlo Tree Search are the decoding schemes used in RL/SL. The 2-OPT is a greedy local search heuristic.

Table 3: Results of SAG + Local Search w.r.t. existing baselines. We evaluate the models on 128 TSP 500 instances and 128 TSP 1000 instances. Baselines include GAT-RL1: Deudon et al. [2018]; GAT-RL2: Kool et al. [2019]; GCN-SL1: Joshi et al. [2019a]; DIMES: Qiu et al. [2022]; DIFUSCO: Sun and Yang [2023] and Att-GCRN Fu et al. [2021].

| Method | Type | TSP500 | | | TSP1000 | | |
| --- | --- | --- | --- | --- | --- | --- | --- |
| | | Length | Gap (%) | Time | Length | Gap (%) | Time |
| Concorde | Solver | 16.5458 | 0.0000 | 37.66m | 23.1182 | 0.0000 | 6.65h |
| Gurobi | Solver | 16.5171 | -0.1733 | 45.63h | - | - | - |
| LKH3 | Heuristic | 16.5463 | 0.0029 | 11.41m | 23.1190 | 0.0036 | 38.09m |
| GAT-RL1 | RL, S | 28.6291 | 73.0293 | 20.18m | 50.3018 | 117.5860 | 37.07m |
| GAT-RL1 | RL, S 2-OPT | 23.7546 | 43.5687 | 57.76m | 47.7291 | 106.4575 | 5.39h |
| GAT-RL2 | RL, S | 22.6409 | 36.8382 | 15.64m | 42.8036 | 85.1519 | 63.97m |
| GAT-RL2 | RL, G | 20.0188 | 20.9902 | 1.51m | 31.1526 | 34.7539 | 3.18m |
| GAT-RL2 | RL, BS | 19.5283 | 18.0257 | 21.99m | 29.9048 | 29.2359 | 1.64h |
| GCN-SL1 | SL, G | 29.7173 | 79.6063 | 6.67m | 48.6151 | 110.2900 | 28.52m |
| GCN-SL1 | SL, BS | 30.3702 | 83.5523 | 38.02m | 51.2593 | 121.7278 | 51.67m |
| GCN-SL1 | SL, BS* | 30.4258 | 83.8883 | 30.62m | 51.0992 | 121.0357 | 3.23h |
| DIMES | RL+S | 18.84 | 13.84 | 1.06m | 26.36 | 14.01 | 2.38m |
| DIMES | RL+MCTS | 16.87 | 1.93 | 2.92m | 23.73 | 2.64 | 6.87m |
| DIMES | RL+AS+MCTS | 16.84 | 1.76 | 2.15h | 23.69 | 2.46 | 4.62h |
| DIFUSCO | SL+MCTS | 16.63 | **0.46** | 10.13m | 23.39 | 1.17 | 24.47m |
| Att-GCRN | SL+RL MCTS | 16.7471 | 1.2169 | 31.17s + 3.33m | 23.5153 | 1.7179 | 43.94s + 6.68m |
| **UTSP (Ours)** | UL, Search | 16.6846 | 0.8394 | 1.37m + 1.33m | 23.3903 | **1.1770** | 3.35m+ 2.67m |

We compare our model with existing solvers as well as different learning-based algorithms. The performance of our method is averaged of four runs with different random seeds. The running time for our method is divided into two parts: the inference time (building the heat map $\mathcal{H}$) and the search time (running search algorithm).

On small instances, our results match the ground-truth solutions and generate average gaps of **-0.00009%**, **-0.002%** and **-0.00011%** respectively on instances with $n = 20, 50, 100$, where the negative values are the results of the rounding problem. The total runtime of our method remains competitive w.r.t. all other learning baselines. On larger instances with $n = 200, 500$ and $1,000$, we notice that traditional solvers (Concorde, Gurobi) fail to generate the optimal solutions within reasonable time when the size of problems grows. For RL/SL baselines, they generate results far away from ideal solutions, particularly for cases with $n = 1,000$. Our UTSP method is able to obtain **0.0918%**, **0.8394%** and **1.1770%** on TSP 200, 500 and 1,000, respectively. We remark that UTSP outperforms the existing learning baselines on larger instances (TSP 200, 500, 1000) [2]. More discussion between Fu et al. [2021] and UTSP can be found in Appendix C.

Our model takes less training time than RL/SL method because we require very few training instances. Taking TSP 100 as an example, RL/SL needs 1 million training instances, and the total training time can take one day using a NVIDIA V100 GPU, while our method only takes about 30 minutes with 2,000 training instances. The training data size does not increase w.r.t. TSP size. Our training data consists of 2,000 instances for TSP 200, 500 and 1,000. At the same time, the UTSP model also remains very lightweight. On TSP 100, we use a 2-layer SAG with 64 hidden units and the model consists of 44,392 trainable parameters. In contrast, RL method in Kool et al. [2019] takes approximately 700,000 parameters and the SL method in Joshi et al. [2022] takes approximately 350,000 parameters.

## 5.3 Expressive Power of GNNs

Our UL method generalizes well to unseen examples without requiring a large number of training samples. This is because the loss function in Equation 2 is fully differentiable w.r.t. the parameters

---

[2]On TSP 500, when we increase the time budget for Search, we achieve 0.42% in 1.37m + 8.32m.

in SAG and we are able to train the model in an end-to-end fashion. In other words, given a heat

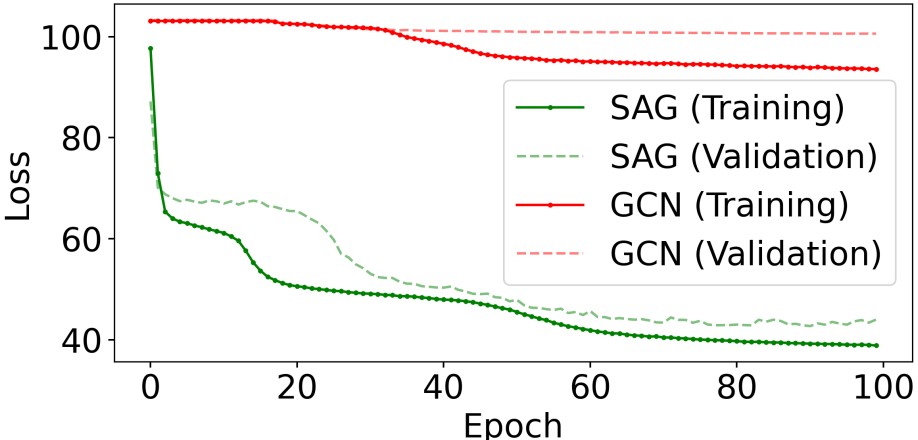

Figure 2: TSP 100 training curve using Unsupervised Learning surrogate loss. We compare two GNN models: GCN Kipf and Welling [2016] and SAG Min et al. [2022], where GCN is a low-pass model and SAG is a low-pass + band-pass model.

map $\mathcal{H}$, the model learns to assign large weights to more promising edges and small weights to less promising ones through backpropagation without any prior knowledge of the ground truth or any exploration step. However, when using SL, the model learns from the TSP solutions, which fails when multiple solutions exist or the solutions are not optimal Li et al. [2018]. While for RL, the model often encounters an exploration dilemma and is not guaranteed to converge Bengio et al. [2021]Joshi et al. [2019b]. Overall, UTSP requires fewer training samples and has better generalization compared to SL/RL models.

We aim to generate a non-smooth soft indicator matrix $\mathbb{T}$ and build an expressive heat map $\mathcal{H}$ to guide the search algorithm. However, most GNNs aggregate information from adjacent nodes and these aggregation steps usually consist of local averaging operations, which can be interpreted as a low-pass filter and causes the oversmoothing problem Wenkel et al. [2022]. The low-pass model generates a smooth soft indicator matrix $\mathbb{T}$, which finally makes the elements in $\mathcal{H}$ become indistinguishable. So it becomes difficult to discriminate whether the edges belong to the optimal solution or not. In our model, we assume all nodes in the graph are connected, so every node has $n-1$ connected to neighbouring nodes. This means every node receives messages from all other nodes and we have a global averaging operation over the graph, this can lead to severe oversmoothing issue.

To avoid oversmoothing, one solution is to use shallow GNNs. However, this would result in narrow receptive fields and create the problem of underreaching Barceló et al. [2020]. Our model uses SAG because this scattering-based method helps overcome the oversmoothing problem by combining band-pass wavelet filters with GCN-type filters Min et al. [2022]. Figure 2 illustrates the training loss on TSP 100 and the differences between our SAG model and the graph convolutional network (GCN) Kipf and Welling [2016], where GCN only performs low-pass filtering on graph signals Nt and Maehara [2019]. When using GCN, the training loss decreases slowly, and the validation loss reaches a plateau after we train the model for 20 epochs. This is because the low-pass model generates a smooth $\mathbb{T}$. Such a smooth $\mathbb{T}$ results in an indistinguishable $\mathcal{H}$, detrimentally impacting the training process. Instead, we observe lower training and validation loss when using SAG; this suggests that SAG generates a more expressive representation which facilitates the training process.

Figure 3 illustrates the generated heat maps using GCN and SAG on a TSP 100 instance, we choose this instance from the validation set randomly. When using the GCN, due to the oversmoothing problem, the model generates a smooth representation and $\mathcal{H}$ becomes indistinguishable. The elements in $\mathcal{H}$ have a small variance and most of them are $\sim 0.01$. Instead, the SAG generates a discriminative representation and the elements in the heat map have a larger variance.

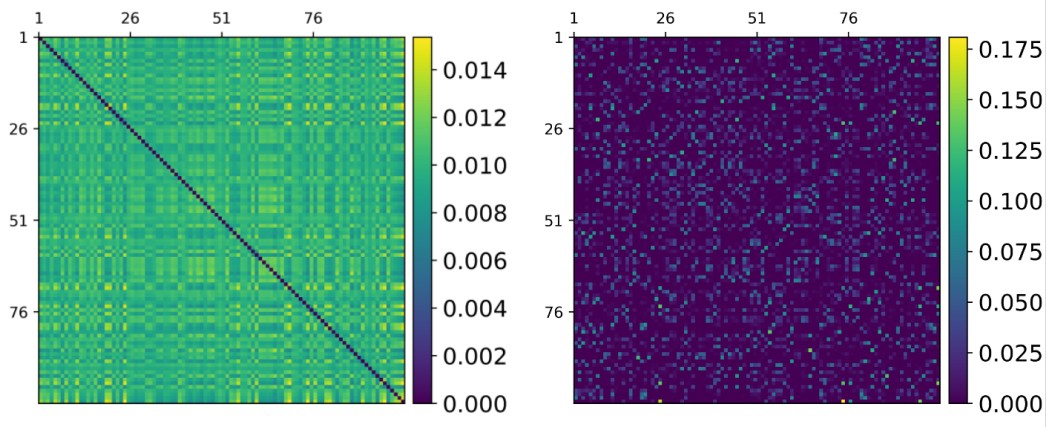

Figure 3: Left: The heat map $\mathcal{H}$ generated using GCN on TSP 100. The diagonal elements are set to 0. $X$-axis and $y$-axis are the city indices, right: The heat map $\mathcal{H}$ generated using SAG on TSP 100. The diagonal elements are set to 0. $X$-axis and $y$-axis are the city indices.

Here, we train both GCN and SAG with the same loss function. So the differences illustrated in Figure 3 are the direct result of overcoming the oversmoothing problem.

## 6 Search Space Reduction

To understand what happens during our training process, we study how the prediction edge set $\Pi$ changes with training time. As mentioned, let $\Pi$ denote undirected edge set in $\mathcal{H}'$, and let $\Gamma$ denote the ground truth edge set, $\eta = |\Gamma \cap \Pi|/|\Gamma|$ is the extent of how good our prediction set $\Pi$ covers the solution $\Gamma$. If $\eta = 1$, then $\Gamma$ is a subset of $\Pi$, which means our prediction edge set successfully covers all ground truth edges. Similarly, $\eta = 0.95$ means we cover 95% ground truth edges.

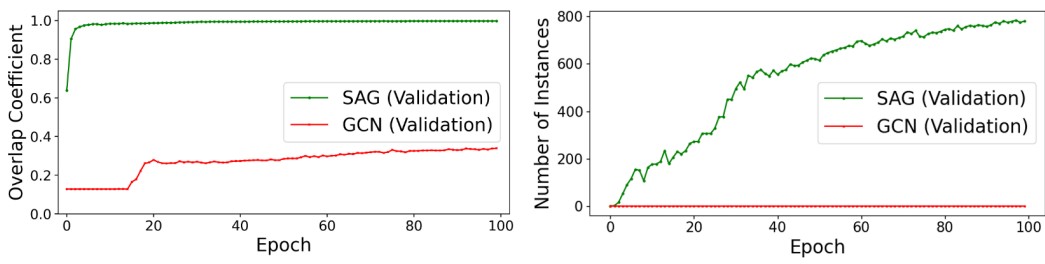

Figure 4: Left:Average edge overlap coefficient $\eta$ w.r.t. training epochs using SAG and GCN on TSP 100 ($M = 10$), right: Number of fully covered instances w.r.t. training epochs using SAG and GCN on TSP 100. The validation set consists of 1,000 samples ($M = 10$).

Figure 4 shows how the average overlap coefficient $\eta$ changes with training epochs. We calculate the coefficient based on 1,000 validation instances in TSP 100. We notice that the coefficient quickly increases to $\sim 98\%$ after we train SAG for 10 epochs. This suggests that the surrogate loss successfully encourages the SAG to put more weights on the more promising edges. We also compare the performance with GCN. Since the loss does not decrease significantly during our training when using GCN (shown in Figure 2), it is not surprising to see the average overlap coefficient of GCN always maintains at a relatively low level. After training the model for 100 epochs, SAG model has an average coefficient of 99.756% while GCN only has 33.893%.

We then study the number of cases where our prediction edge set $\Pi$ covers the ground truth solution. Figure 4 (right) illustrates how the number of fully covered instances ($\eta = 1$) changes with time.

After training the model for 100 epochs, we observe 780 fully covered instances in 1,000 validation samples using SAG while 0 instances using GCN. Finally, we calculate the average of size $|\Pi|$. Our results show that SAG has an average size of $583.134$ edges, while for GCN, the number is $738.739$.

These results also indicate a correspondence between the loss and the quality of our prediction. In most SL tasks such as classification or regression tasks, a smaller validation loss usually means we achieve better performance and the minimum of the loss corresponds to the optimal solution (100% accuracy). However, there is no theoretical guarantee that our loss in Equation 2 also measures the solution quality. Our empirical results demonstrate that a lower surrogate loss encourages the model to assign larger weights on the promising edges and reduces the search space. This implies that we can assess the quality of the generated heat maps using our loss in Equation 2.

Overall, the UL training reduces the search space from $4950$ edges to $583.134$ edges with over $99\%$ overlap accuracy on average. This helps explain why our search algorithm is able to perform well within reasonable search time.

## 7  Conclusion

In this paper, we propose UTSP, an Unsupervised Learning method to solve the TSP. We build a surrogate loss that encourages the GNN to find the shortest path and satisfy the constraint that the path should be a Hamiltonian Cycle. The surrogate loss function does not rely on any labelled ground truth solution and helps alleviate sparse reward problems in RL. UTSP uses a two-phase strategy. We first build a heat map based on the GNN's output. The heat map is then fed into a search algorithm. Compared with RL/SL, our method vastly reduces training cost and takes fewer training samples. We further show that our UL training helps reduce the search space. This helps explain why the generated heat maps can guide the search algorithm. On the model side, our results indicate that a low-pass GNN will produce an indistinguishable representation due to the oversmoothing issue, which results in unfavorable heat maps and fails to reduce the search space. Instead, after incorporating band-pass operators into GNN, we can build efficient heat maps that successfully reduce search space. Our findings show that the expressive power of GNNs is critical for generating a non-smooth representation that helps find the solution.

In conclusion, UTSP is competitive with or outperforms other learning-based TSP heuristics in terms of solution quality and running speed. In addition, UTSP takes $\sim 10\%$ of the number of parameters and $\sim 0.2\%$ of (unlabelled) training samples, compared with RL or SL methods. Our UTSP framework demonstrates that by providing a surrogate loss and a GNN which encourages a non-smooth representation, we can learn the hidden patterns in TSP instances without supervision and further reduce the search space. This allows us to build a heuristic by exploiting a small amount of unlabelled data. Future directions include designing more expressive GNNs (such as adding edge features) and using different surrogate loss functions. We anticipate that these concepts will extend to more combinatorial problems.

## 8  Acknowledgement

This project is partially supported by the Eric and Wendy Schmidt AI in Science Postdoctoral Fellowship, a Schmidt Futures program; the National Science Foundation (NSF) and the National Institute of Food and Agriculture (NIFA); the Air Force Office of Scientific Research (AFOSR); the Department of Energy; and the Toyota Research Institute (TRI).

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

# A Discussion

## A.1 Definition of heat map

We can write $\mathcal{H}$ as:

$$\mathcal{H} = \mathbb{T}\mathbb{V}\mathbb{T}^T, \qquad (3)$$

where

$$\mathbb{V} = \begin{pmatrix} 0 & 1 & 0 & 0 & \cdots & 0 & 0 & 0 \\ 0 & 0 & 1 & 0 & \cdots & 0 & 0 & 0 \\ 0 & 0 & 0 & 1 & \cdots & 0 & 0 & 0 \\ \vdots & \vdots & \vdots & \ddots & \ddots & \vdots & \vdots & \vdots \\ 0 & 0 & 0 & 0 & \ddots & 1 & 0 & 0 \\ 0 & 0 & 0 & 0 & \cdots & 0 & 1 & 0 \\ 0 & 0 & 0 & 0 & \cdots & 0 & 0 & 1 \\ 1 & 0 & 0 & 0 & \cdots & 0 & 0 & 0 \end{pmatrix}$$

is the Sylvester shift matrix Sylvester [1909], $\mathbb{V} \in \mathbb{R}^{n \times n}$. We can interpret $\mathbb{V}$ as a cyclic permutation operator that performs a circular shift.

## A.2 Unsupervised Loss

We can also write Equation 2 in a more compact form:

$$\mathcal{L} = \lambda_1 \underbrace{\sum_{i=1}^{n}(\sum_{j=1}^{n} \mathbb{T}_{i,j} - 1)^2}_{\text{Row-wise constraint}} + \underbrace{\sum_{i=1}^{n}\sum_{j=1}^{n} \tilde{\mathbf{D}}_{i,j}\mathcal{H}_{i,j}}_{\text{Minimize the distance}}, \qquad (4)$$

where $\tilde{\mathbf{D}} = \mathbf{D} + \lambda_2 \mathcal{I}_n$, $\mathcal{I}_n \in \mathbb{R}^{n \times n}$ is the identity matrix.

Table 4: Search parameters for all the TSP experiments.

| | $\alpha$ | $\beta$ | $M$ | $K$ | $T$ |
|---|---|---|---|---|---|
| TSP-20 | 0 | 10 | 8 | 10 | 60 |
| TSP-50 | 0 | 10 | 8 | [5, 15) | 150 |
| TSP-100 | 0 | 10 | 8 | [5, 35) | 300 |
| TSP-200 | 0 | 10 | 8 | [10, 90) | 600 |
| TSP-500 | 0 | 50 | 5 | [30, 130) | 1000 |
| TSP-1000 | 0 | 50 | 5 | [10, 110) | 2000 |

# B Training and Search Details

We train our model using Adam Kingma and Ba [2014]. All models are trained using Nvidia V100 GPU. All the search-related parameters are listed in Table 4. $M$ is the size of the candidate set of each city. $K$ is the maximal number of edges we can remove in one action, and for each round of local search, we randomly select one number from the listed interval. $T$ is the total number of actions we will try to expand one node. Here, we set $\alpha = 0$ to show that our unsupervised model generates an informative heat map. Lower $\alpha$ means the local search algorithm focuses more on the edges with higher heat map value. Actually, in the experiments, we find the results are similar with $\alpha \leq 1$.

# C Running Time Discussion

As discussed in Kool et al. [2019], running time is important but hard to compare since it is affected by many factors. In the table 1, we report the time for solving all the test instances.

For the UTSP (our method) and the state-of-the-art learning-based method Att-GCRN Fu et al. [2021], we run the search algorithm on exactly the same environment (one Intel Xeon Gold 6326) for a fair comparison. And for other baselines, we refer to the results from Fu et al. [2021]. So, the time there is only for indicative purposes since the computing hardware is different.

## D  Proof

**Lemma D.1.** *Let $q_i$ denote the row index of the non-zero element in $i$-th column in $\mathbb{T}$, $\mathbb{T}_{q_i,i} = 1$, $q_i \in \{1, 2, 3, 4, ..., n\}$. When each row and column in $\mathbb{T}$ have one value 1 (True) and $n - 1$ value 0 (False), then $q_i = q_j$ if and only if $i = j$.*

*Proof.* If there exist a $(i, j)$ pair where $q_i = q_j$ when $i \neq j$, then $\mathbb{T}_{q_i,i} = 1$ and $\mathbb{T}_{q_j,j} = \mathbb{T}_{q_i,j} = 1$. This means $q_i$-th row has two non-zero elements, which leads to a contradiction. $\square$

**Lemma D.2.** *Consider graph $\mathcal{G}$ with $n$ nodes, for any $\mathbb{T} \in \mathbb{R}^{n \times n}$ with $\mathbb{T}_{i,j} \geq 0$, when each row and column in $\mathbb{T}$ have one value 1 (True) and $n - 1$ value 0 (False). Then, each row and column in $\mathcal{H}$ also have one value $1$ (True) and $n - 1$ value $0$ (False), which means each city corresponds to only one beginning point and one ending point.*

*Proof.* First, it is clear that for $\forall a, b, \mathcal{H}_{a,b} \in \mathbb{Z}_{\geq 0}$. Let's assume $\mathbb{T}_{a,l} = \mathbb{T}_{b,m} = 1 (a \neq b, l \neq m)$, since $\mathcal{H}_{i,j} = \sum_{k=1}^{n} \mathbb{T}_{i,k} \mathbb{T}_{j,k+1 (\mathrm{mod}\ n)}$, we then have

$$\sum_{j=1}^{n} \mathcal{H}_{a,j} = \sum_{j=1}^{n} \sum_{k=1}^{n} \mathbb{T}_{a,k} \mathbb{T}_{j,k+1 (\mathrm{mod}\ n)}$$
$$= \sum_{k=1}^{n} \mathbb{T}_{a,k} \{ \sum_{j=1}^{n} \mathbb{T}_{j,k+1 (\mathrm{mod}\ n)} \}$$
$$= \sum_{k=1}^{n} \mathbb{T}_{a,k} = 1.$$

This implies that the summation of each row in $\mathcal{H}$ is 1. Similarly,

$$\sum_{i=1}^{n} \mathcal{H}_{i,b} = \sum_{i=1}^{n} \sum_{k=1}^{n} \mathbb{T}_{i,k} \mathbb{T}_{b,k+1 (\mathrm{mod}\ n)}$$
$$= \sum_{k=1}^{n} \mathbb{T}_{b,k+1 (\mathrm{mod}\ n)} \{ \sum_{i=1}^{n} \mathbb{T}_{i,k} \}$$
$$= \sum_{k=1}^{n} \mathbb{T}_{b,k+1 (\mathrm{mod}\ n)} = 1.$$

This suggests the summation of each column in $\mathcal{H}$ is 1. Since each element in $\mathcal{H}_{i,j} \in \mathbb{Z}_{\geq 0}$, we can then conclude that each row and column in $\mathcal{H}$ have one value 1 (True) and $n - 1$ value 0 (False). Also, $\mathcal{H}_{ii} = \sum_{k=1}^{n} \mathbb{T}_{i,k+1 (\mathrm{mod}\ n)} \mathbb{T}_{i,k} = 0$, this indicates that the elements in the main diagonal of $\mathcal{H}$ are 0, which implies no self-loops.

As mentioned, the $i$-th row in $\mathcal{H}$ is the probability distribution of directed edges start from city $i$, and the $j$-th column is the probability distribution of directed edges end in city $j$. Because each row and column in $\mathcal{H}$ have one value 1 (True) element and $\mathcal{H}$'s diagonal entries are all zero, this means that each city is the beginning point of one directed edge and is also the ending point of another different directed edge. $\square$

**Lemma D.3.** *There is at least one cycle in $\mathcal{H}$ which contains $n$ edges and visits all cities when each row and column in $\mathbb{T}$ have one value 1 (True) and $n - 1$ value 0 (False).*

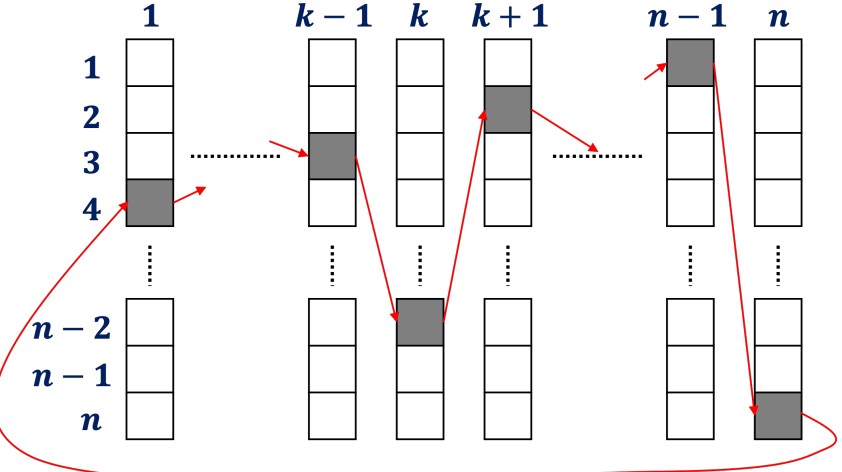

Figure 5: Illustration of building a cycle from transition matrix $\mathbb{T}$. Here $q_{k-1} = 3$, $q_k = n - 2$, $q_{k+1} = 2$, $q_{n-1} = 1$, $q_n = n$ and $q_1 = 4$, this means $\mathcal{H}$ contains the following four directed edges: $3 \rightarrow n - 2$, $n - 2 \rightarrow 2$, $1 \rightarrow n$ and $n \rightarrow 4$.

*Proof.* **Lemma** D.1 indicates that $q_i \neq q_j$ when $i \neq j$ and $q_i \in \{1, 2, 3, 4, ..., n\}$, then $\cup_{i=1}^{n} q_i = \{1, 2, 3, 4, ..., n\}$. From Equation 3, we have

$$\begin{aligned}
\mathcal{H}_{q_i, q_{i+1}} &= \sum_{k=1}^{n} \mathbb{T}_{q_i, k} \mathbb{T}_{q_{i+1}, k+1 (\text{mod } n)} \\
&\geq \mathbb{T}_{q_i, i} \mathbb{T}_{q_{i+1}, i+1 (\text{mod } n)} \\
&\geq 1.
\end{aligned}$$

Using **Lemma** D.2, since each row and column in $\mathcal{H}$ have one value 1 (True) and $n - 1$ value 0 (False), it suffices to show that $1 \geq \mathcal{H}_{q_i, q_{i+1}} \geq 1$, therefore $\mathcal{H}_{q_i, q_{i+1}} = 1$. This suggests that there is a directed edge from city $q_i$ to $q_{i+1}$. We can then construct a cycle $\mathcal{C}$ from $\mathcal{H}$, we can write $\mathcal{C}$ as

$$q_1 \rightarrow q_2 \rightarrow q_3 \rightarrow q_4 \rightarrow ... \rightarrow q_n \rightarrow q_1,$$

where $\rightarrow$ is a directed edge. Since $\cup_{i=1}^{n} q_i = \{1, 2, 3, 4, ..., n\}$, cycle $\mathcal{C}$ visits all $n$ cities and have $n$ edges. One example of how to build $\mathcal{H}_{q_i, q_{i+1}}$ from $\mathbb{T}$ is shown in Figure 5. $\square$

**Corollary D.4.** *$\mathcal{H}$ represents one Hamiltonian Cycle when each row and column in $\mathbb{T}$ have one value 1 (True) and $n - 1$ value 0 (False).*

*Proof.* From **Lemma** D.3, cycle $\mathcal{C}$ contains $n$ edges and visits all cities, if there exists another edge $(i, j)$ which does not belong to $\mathcal{C}$, then city $i$ is the starting point of at least two edges and city $j$ is the ending point of at least two edges. This results in a contradiction with **Lemma** D.2. Thus, it suffices to conclude that $\mathcal{C}$ visits each city exactly once and $\mathcal{H}$ only contains the edges in $\mathcal{C}$. This implies that $\mathcal{H}$ represents one Hamiltonian Cycle. $\square$

