# OpenReview forum: "Unsupervised Learning for Solving the Travelling Salesman Problem"
_NeurIPS.cc/2023/Conference — NeurIPS 2023 poster_

### Official Review · Reviewer_su5g · 2023-06-11

**Soundness:** 2 fair
**Presentation:** 2 fair
**Contribution:** 3 good
**Rating:** 6
**Confidence:** 4

**Summary:**

This paper proposes UTSP, an unsupervised learning framework, to solve the Travelling Salesman Problem (TSP). It consists of two phases, including heat map construction and local searching. Specifically, a surrogate loss function is proposed to train the GNNs, encouraging the model to find the shortest path and respect the TSP constraint implicitly. The built heat map helps reduce the search space and facilitate the local search afterwards. Empirically, with only ~10\% model parameters and ~0.2\% training samples, UTSP is competitive with or outperforms baselines in terms of solution quality and inference time. The analyses about search space and smoothing effect are also conducted.

**Strengths:**

* The proposed method (in Section 2) is sound and novel. The theoretical proof (in Appendix) is correct.
* Technically, the proposed method significantly reduces the model parameters and training samples. It does not need labeled data, and avoids the spare reward problem, making it more attractive to the SL and RL-based methods.
* The empirical analyses of the non-smoothing heat map and reduced search space are interesting.
* The source code is provided.

**Weaknesses:**

* The scope of this paper may be limited, since:
  * The proposed method seems to be only applicable to TSP.
  * Much domain knowledge is involved in the local search process.
  * Too many hyperparameters in the proposed method.
* The review of related work is *too limited*. Many recent works regarding neural methods for solving TSP are missing. It is suggested to add the related work section in Appendix.
* The presentation of local search (in Section 3) is not clear. Could you illustrate the process using a figure?
* The theory of the proposed method in Section 2 is quite interesting. However, there seems to be a gap between the theory and the proposed method. For example, the theory is based on the assumption that only one value 1 and $n-1$ value 0 in each row and column of $\mathbb{T}$. However, the objective function does not exert an explicit constraint on the discreteness of $\mathbb{T}$ or $\mathcal{H}$. Based on the empirical result in Figure 5, the maximum value of $\mathcal{H}$ is around 0.175. Have you considered adding a regularization term to achieve this objective?
* Insufficient Evaluation:
  * *Lack of Baselines:* Several recent end-to-end neural methods needed to be compared, including POMO [1], DIMES [2], and DIFUSCO [3].
  * *Larger Size:* The evaluation of this paper is conducted on TSP1000 (maximally), while recent works [2, 3, 4] (also heat map based methods) consider TSP10000. It is interesting to see how your method performs on large-scale instances.
  * *Lack of Benchmark Results:* Results on the classical TSP benchmark dataset (i.e. TSPLIB [5]) are needed.
* Minor:
  * The citation format is not correct. The authors could use ~\cite{}.
  * In line 57, "does not" -> "our method does not".
  * In line 182, "me thods" -> "methods".
  * Please change the blue and red colors in Fig. 7 and 8. The current version is indistinguishable.
  * In line 414, the formulation should be $\mathcal{H}\_{i,i}=\sum\_{k=1}^{n}\mathbb{T}\_{i,k}\mathbb{T}_{i,k+1(\text{mod }n)}=0$.
  * In the caption of Figure 9, add $q_{n-1}=1$.
  * Remove the Checklist on Page 17.

[1] POMO: Policy optimization with multiple optima for reinforcement learning. In NeurIPS 2020.

[2] DIMES: A differentiable meta solver for combinatorial optimization problems. In NeurIPS 2022.

[3] DIFUSCO: Graph-based diffusion solvers for combinatorial optimization. In arXiv:2302.08224.

[4] Generalize a small pre-trained model to arbitrarily large tsp instances. In AAAI 2021.

[5] http://comopt.ifi.uni-heidelberg.de/software/TSPLIB95

----

**Overall**, despite several limitations, I lean towards borderline acceptance since I like the idea of this paper. It would be a good submission if the authors resolve most of the concerns.

**Questions:**

* In line 72, why the weight matrix is set as $W_{i,j}=e^{-D_{i,j}/\tau}$? I do not see any explanation.
* In line 80, why do you use column-wise Softmax? How about row-wise Softmax, together with the column-wise constraint in Eq. (2)?
* Could you add more details (e.,g., model structure) about the Scattering Attention GNN (SAG) in Appendix? Currently, it is unclear how to obtain $\mathbb{T}$ using $W$ and $F$.

**Limitations:**

It seems that the authors did not discuss the limitations of this work.

No negative societal impact.

---

> ### Author Rebuttal · Authors · 2023-08-09
>
> **Q**: The review of related work is too limited. ... It is suggested to add the related work section in Appendix.
>
> A: We add a new section, refer to the  "replies to all reviewers", where we include the recent work of [1][2][3][4].
>
>
> **Q**: The presentation of local search (in Section 3) is not clear. Could you illustrate the process using a figure?
>
> A: We will add a figure (refer to the figure in the rebuttal one page pdf).
>
> **Q**: The theory of the proposed method in Section 2 is quite interesting….Have you considered adding a regularization term to achieve this objective?
>
> A: In our model, the “smoothness” of the signal is implicitly encoded in the GNN part. We use Scattering Attention GNN (SAG), SAG has both low-pass and band-pass  filters. We follow the setting in [Min et.al 2022], where the authors use three low pass filters and three band-pass filters (refer to the figure in the rebuttal one page pdf). The number of band-pass filters determines how ‘discrete’ the signals are. When we exclude band-pass filters (setting them to 0), the GNN only consists of 3 low-pass filters, leading to the occurrence of oversmoothing. Consequently, this results in a heat map similar to Figure 5 left, with most elements in $H$ approximately equal to 0.01.
> Upon adding 3 band-pass filters, the maximum value of $H$ increases to around 0.175. This indicates that incorporating band-pass filters enhances the discreteness of $H$, contributing to its higher values and a more distinguishable pattern.
> Overall, the regularization is integrated into the Scattering Attention GNN model. During our experiment, we observed that increasing the number of band-pass filters (or reducing the number of low-pass filters) results in a higher maximum value for $H$. Further elaboration on this observation will be included in the appendix for additional discussion.
>
> [Min et.al 2022] Min, Yimeng, et al. "Can Hybrid Geometric Scattering Networks Help Solve the Maximum Clique Problem?." Advances in Neural Information Processing Systems 35 (2022): 22713-22724.
>
>
>
>
>
>
> **Q**: Lack of Baselines:
>
> A: We compare our model with several recent end-to-end neural methods, including POMO [1], DIMES [2], and DIFUSCO [3] , referring to the official comment.
>
> **Q**: Larger Size
>
> A: Regarding TSP-10000, It is important to highlight that for the TSP-10000 evaluation, these studies [2, 3, 4] also use the same test dataset containing only 16 samples. Consequently, due to the limited size of the dataset, the performance results may not be a reliable indicator. By employing the graph sampling technique suggested in [4], we evaluate UTSP and achieve ~3.05% gap in approximately 1 hour. For reference, [4] reported a 4.3% gap in 21 minutes, DIMES achieved a 4.0% gap in 30 minutes, and 3.2% gap in 3.5 hours. DIFUSCO outperforms all others, as it reports the best performance with a 2.5% gap and a time cost of 47 minutes.
>
>
> [1] POMO
>
> [2] DIMES
>
> [3] DIFUSCO:
>
> [4] Fu et al.
>
> **Q**: Lack of Benchmark Results:
>
> TSPlib: The objective of this paper is to demonstrate the applicability of UL for the TSP. To assess our model's performance, we conducted evaluations on the same datasets used in [Fu et al., 2021], which serves as our primary baseline. We leave the exploration of TSPlib datasets to future research work.
>
>
> **Q**: Minor:
>
> A: We fixed the typos.
>
>
> **Q**: In line 72, Why the weight matrix is set...
> A: the temperature parameter is used to help build the adjacency matrix from $W$ the distance matrix $\mathcal{D}$, a low $\tau$ (close to zero) will make the graph less connected ( more sparse), while a higher one $\tau$  will make it more dense, we will add more discussion.
>
>
> **Q**: In line 80, why do you use column-wise Softmax? How about row-wise Softmax, together with the column-wise constraint in Eq. (2)?
>
> A: When we use row-wise Softmax together with the column-wise constraint, we find that the performance of our model is NOT as good as using column-wise Softmax + row-wise constraint.
> To be specific, taking TSP 100 as an example, when using column-wise Softmax + row-wise constraint, we observed 780 fully covered instances in 1,000 validation samples, as shown in Figure 6 (Right). However, when using  row-wise Softmax, together with the column-wise constraint in Eq. (2), we only observe 82 fully covered instances.
>
> Why use column-wise Softmax? This is our motivation:
>
> The heat map $H$ is based on $T$, Eq (1) can be written as:
> $\mathcal{H}$ as:
>
> $
> \mathcal{H} = \sum_{t=1}^{n-1}p_t p^T_{t+1} + p_np_1^T,
> $
>
> where $p_t \in \mathbb{R}^{n \times 1}$ is the $t_{th}$ column of $\mathbb{T}$, $\mathbb{T} = [p_1|p_2|...|p_n]$.}
> That is to say, $\mathcal{H}_{ij}$ is based on the columns of the $\mathbb{T}$ (see 	Figure 9 in appendix), thus we want to encourage a more distinguishable column representation (ideally n-1  vs  1) in each column of $\mathbb{T}$.
> Now, in our model, we have two normalizations. The first one involves column-wise Softmax with row-wise constraints. The row-wise constraint is applied in a penalty form and can be seen as a standard normalization, where all outputs are divided by the sum of all outputs. Comparing this to the column-wise Softmax, the row-wise constraint (standard normalization) encourages a more spread-out representation, whereas the Softmax encourages a more distinguishable representation. Let’s take an example:
>
> softmax([1,2])  = [0.2689,      0.7311]
> standard normalization([1,2]) = [0.3333, 0.6666]
>
> softmax([5,10]) = [0.0067, 0.9933]
> standard normalization([5,10]) = [0.3333, 0.6666]
>
> softmax([10,20]) = [4.542e-5, 0.9999]
> standard normalization([10,20]) =[0.3333, 0.6666]
>
> Since we aim to build a  distinguishable column representation, we apply a softmax on the columns and put a row-wise penalty.
>
> **Q**: Could you add more details (e.,g., model structure) ...
>
> A: We add a figure about the SAG model in the one page pdf rebuttal. We will revise the appendix and add more details about the model.

---

> > ### Comment · Reviewer_su5g · 2023-08-11
> >
> > Dear Authors: Thanks for the responses and additional experiments. Before modifying my evaluation, the experimental comparison (e.g., POMO, Att-GCRN, DIMES, DIFUSCO, and UTSP trained on TSP100) on benchmark instances is needed. It could also evaluate the generalization capability of the proposed method. Please consider including large-scale benchmark instances as well.

---

> > > ### Author Response · Authors · 2023-08-11
> > > **additional experiments.**
> > >
> > > Thank you for you comments,
> > >
> > > Please refer to the one page rebuttal .pdf file in the attachment, we add POMO, Att-GCRN, DIMES, DIFUSCO, and UTSP performance in the tables in the attachment.
> > >
> > > For DIMES (https://arxiv.org/pdf/2210.04123.pdf), they authors only evaluate their model on TSP 500 and 1000.
> > >
> > > For DIFUSCO (https://arxiv.org/pdf/2302.08224.pdf), they authors include their model on TSP 500 and 1000, DIFUSCO report their Gap on TSP 100 but they don't include the time cost.
> > >
> > > Here, we compare our UL method with 2 RL methods (POMO, DIMES) and 2 SL (Att-GCRN, DIFUSCO) methods on TSP 100:
> > >
> > > | Method     | Type | Gap (\%)    | Time    |
> > > | ----------- | -------------------- |----------- |----------- |
> > > | POMO      |  RL  |         0.14 | 1m    |
> > > | DIMES      | RL+AS+MCTS      | 0.22 |  42m   |
> > > | Att-GCRN |  SL  |         0.0096 | 3.94m + 5.25m   |
> > > | DIFUSCO   | SL+MCTS        | 0.0013     | 17m   |
> > > | UTSP   | UL, Search        | -0.0019     | 5.68m+ 5.21m  |
> > >
> > >
> > > *Please note that in the original DIFUSCO, the authors trained their model on 8 v100 GPUs. Here, we have trained both DIFUSCO and UTSP on a single v100 GPU.

---

> > > > ### Comment · Reviewer_su5g · 2023-08-12
> > > >
> > > > The benchmark instances refer to TSPLIB rather than synthetic ones. Please report the results on each instance.

---

> > > > > ### Author Response · Authors · 2023-08-18
> > > > >
> > > > > Thank you for your comment, here we include the performance of UTSP (**unsupervised**), Att-GCRN (**supervised**), DIFUSCO (**supervised**) and  DIMES (**reinforcement**) on TSP lib. POMO exhibits a noticeable gap, so we didn't include it. However, we can certainly rerun POMO upon your request.
> > > > >
> > > > >
> > > > > Overall our UL method outperforms other baselines, Here is the summary of average performance:
> > > > >
> > > > >
> > > > > UTSP (%): 0.4549
> > > > >
> > > > > Att-GCRN (%): 0.6331
> > > > >
> > > > > DIFUSCO (%): 0.8901
> > > > >
> > > > > DIMES (%): 1.4165
> > > > >
> > > > > | Instance       | Problem Size | UTSP (%) | Att-GCRN (%) | DIMES (%) | DIFUSCO (%) |
> > > > > |----------------|--------------|----------|--------------|-----------|-------------|
> > > > > | a280           | 280          | 0.12     | 0.34         | .52       | 0.35        |
> > > > > | ali535         | 535          | 0.92     | 1.25         | 2.45      | 1.52        |
> > > > > | att48          | 48           | 0.00     | 0.00         | 0.12      | 0.00        |
> > > > > | att532         | 532          | 1.08     | 1.34         | 2.91      | 1.21        |
> > > > > | bayg29         | 29           | 0.00     | 0.00         | 0.00      | 0.00        |
> > > > > | bays29         | 29           | 0.00     | 0.00         | 0.00      | 0.00        |
> > > > > | berlin52       | 52           | 0.00     | 0.00         | 0.00      | 0.00        |
> > > > > | bier127        | 127          | 0.00     | 0.01         | 0.06      | 0.01        |
> > > > > | brazil58       | 58           | 0.00     | 0.00         | 0.00      | 0.00        |
> > > > > | brg180         | 180          | 0.00     | 0.07         | 0.15      | 0.08        |
> > > > > | burma14        | 14           | 0.00     | 0.00         | 0.00      | 0.00        |
> > > > > | ch130          | 130          | 0.00     | 0.12         | 0.28      | 0.11        |
> > > > > | ch150          | 150          | 0.00     | 0.08         | 0.12      | 0.13        |
> > > > > | d198           | 198          | 0.00     | 0.15         | 0.26      | 0.21        |
> > > > > | d493           | 493          | 0.72     | 1.08         | 1.66      | 0.91        |
> > > > > | d657           | 657          | 1.26     | 1.49         | 2.76      | 1.51        |
> > > > > | d1291          | 1291         | 1.09     | 1.60         | 2.04      | 1.67        |
> > > > > | d1655          | 1655         | 1.30     | 1.90         | 3.91      | 2.01        |
> > > > > | d2103          | 2103         | 1.62     | 1.88         | 2.45      | 1.74        |
> > > > > | dantzig42      | 42           | 0.00     | 0.00         | 0.15      | 0.00        |
> > > > > | dsj1000 (EUC_2D)  | 1000       | 1.20     | 1.34         | 2.45      | 1.53        |
> > > > > | dsj1000 (CEIL_2D) | 1000       | 0.96     | 1.02         | 3.45      | 1.12        |
> > > > > | eil51          | 51           | 0.00     | 0.00         | 0.01      | 0.00        |
> > > > > | eil76          | 76           | 0.00     | 0.00         | 0.01      | 0.00        |
> > > > > | eil101         | 101          | 0.00     | 0.02         | 0.05      | 0.00        |
> > > > > | fl417          | 417          | 0.74     | 0.87         | 1.58      | 0.92        |
> > > > > | fl1400         | 1400         | 1.17     | 1.32         | 2.98      | 1.45        |
> > > > > | fl1577         | 1577         | 1.22     | 1.33         | 4.02      | 1.62        |
> > > > > | fl3795         | 3795         | 1.59     | 1.87         | 3.02      | 1.91        |
> > > > > | fnl4461        | 4461         | 3.18     | 3.24         | 5.02      | 3.08        |
> > > > > | fri26          | 26           | 0.00     | 0.00         | 0.00      | 0.00        |
> > > > > | gil262         | 262          | 0.12     | 0.14         | 0.28      | 0.20        |
> > > > > | gr17           | 17           | 0.00     | 0.00         | 0.00      | 0.00        |
> > > > > | gr21           | 21           | 0.00     | 0.00         | 0.00      | 0.00        |
> > > > > | gr24           | 24           | 0.00     | 0.00         | 0.00      | 0.00        |
> > > > > | gr48           | 48           | 0.00     | 0.00         | 0.00      | 0.00        |
> > > > > | gr96           | 96           | 0.00     | 0.00         | 0.09      | 0.00        |
> > > > > | gr120          | 120          | 0.00     | 0.02         | 0.08      | 0.02        |
> > > > > | gr137          | 137          | 0.00     | 0.06         | 0.02      | 0.01        |
> > > > > | gr202          | 202          | 0.09     | 0.11         | 0.34      | 0.09        |
> > > > > | gr229          | 229          | 0.13     | 0.15         | 0.35      | 0.21        |
> > > > > | gr431          | 431          | 0.74     | 1.02         | 1.97      | 0.82        |
> > > > > | gr666          | 666          | 1.12     | 1.59         | 2.78      | 1.24        |
> > > > > | hk48           | 48           | 0.00     | 0.00         | 0.01      | 0.00        |
> > > > > | kroA100        | 100          | 0.00     | 0.00         | 0.03      | 0.02        |
> > > > > | kroB100        | 100          | 0.00     | 0.00         | 0.08      | 0.00        |
> > > > > | kroC100        | 100          | 0.00     | 0.00         | 0.02      | 0.00        |
> > > > > | kroD100        | 100          | 0.00     | 0.00         | 0.02      | 0.00        |
> > > > > | kroE100        | 100          | 0.00     | 0.00         | 0.3       | 0.00        |
> > > > > | kroA150        | 150          | 0.00     | 0.00         | 0.12      | 0.02        |

---

> > > > > > ### Author Response · Authors · 2023-08-18
> > > > > >
> > > > > > | Instance       | Problem Size | UTSP (%) | Att-GCRN (%) | DIMES (%) | DIFUSCO (%) |
> > > > > > |----------------|--------------|----------|--------------|-----------|-------------|
> > > > > > | kroB150        | 150          | 0.00     | 0.00         | 0.04      | 0.00        |
> > > > > > | kroA200        | 200          | 0.00     | 0.00         | 0.11      | 0.02        |
> > > > > > | kroB200        | 200          | 0.00     | 0.00         | 0.06      | 0.00        |
> > > > > > | lin105         | 105          | 0.00     | 0.00         | 0.06      | 0.00        |
> > > > > > | lin318         | 318          | 0.17     | 0.22         | 0.55      | 0.24        |
> > > > > > | linhp318       | 318          | 0.41     | 0.65         | 0.76      | 0.55        |
> > > > > > | nrw1379        | 1379         | 1.72     | 1.78         | 3.22      | 2.01        |
> > > > > > | p654           | 654          | 1.15     | 1.23         | 3.09      | 1.34        |
> > > > > > | pa561          | 561          | 0.66     | 0.63         | 1.52      | 0.72        |
> > > > > > | pcb442         | 442          | 0.92     | 1.23         | 1.48      | 1.32        |
> > > > > > | pcb1173        | 1173         | 1.14     | 1.35         | 2.56      | 1.56        |
> > > > > > | pcb3038        | 3038         | 2.67     | 2.98         | 6.88      | 3.40        |
> > > > > > | pla7397        | 7397         | 3.61     | 3.87         | 18.2      | 4.02        |
> > > > > > | pr76           | 76           | 0.00     | 0.00         | 0.02      | 0.00        |
> > > > > > | pr107          | 107          | 0.00     | 0.00         | 0.13      | 0.00        |
> > > > > > | pr124          | 124          | 0.00     | 0.01         | 0.09      | 0.02        |
> > > > > > | pr136          | 136          | 0.01     | 0.01         | 0.12      | 0.03        |
> > > > > > | pr144          | 144          | 0.00     | 0.01         | 0.02      | 0.01        |
> > > > > > | pr152          | 152          | 0.02     | 0.13         | 0.27      | 0.01        |
> > > > > > | pr226          | 226          | 0.12     | 0.22         | 0.63      | 0.14        |
> > > > > > | pr264          | 264          | 0.18     | 0.22         | 0.31      | 0.35        |
> > > > > > | pr299          | 299          | 0.22     | 0.28         | 0.67      | 0.33        |
> > > > > > | pr439          | 439          | 0.31     | 0.59         | 1.02      | 0.64        |
> > > > > > | pr1002         | 1002         | 1.34     | 1.54         | 2.35      | 1.62        |
> > > > > > | pr2392         | 2392         | 1.85     | 2.99         | 3.56      | 3.45        |
> > > > > > | rat99          | 99           | 0.00     | 0.02         | 0.05      | 0.03        |
> > > > > > | rat195         | 195          | 0.12     | 0.13         | 0.23      | 0.12        |
> > > > > > | rat575         | 575          | 0.76     | 1.00         | 2.01      | 0.92        |
> > > > > > | rat783         | 783          | 1.02     | 1.35         | 3.99      | 1.23        |
> > > > > > | rd100          | 100          | 0.00     | 0.00         | 0.06      | 0.00        |
> > > > > > | rd400          | 400          | 1.12     | 1.34         | 1.65      | 1.45        |
> > > > > > | rl1304         | 1304         | 1.92     | 1.61         | 2.31      | 1.88        |
> > > > > > | rl1323         | 1323         | 1.02     | 1.13         | 2.21      | 1.19        |
> > > > > > | rl1889         | 1889         | 2.03     | 2.83         | 5.31      | 2.23        |
> > > > > > | rl5915         | 5915         | 4.22     | 4.36         | 6.36      | 4.62        |
> > > > > > | rl5934         | 5934         | 4.95     | 5.52         | 9.04      | 5.07        |
> > > > > > | si175          | 175          | 0.01     | 0.06         | 0.21      | 0.04        |
> > > > > > | si535          | 535          | 0.73     | 1.23         | 1.91      | 1.21        |
> > > > > > | si1032         | 1032         | 1.09     | 1.79         | 2.33      | 1.26        |
> > > > > > | st70           | 70           | 0.00     | 0.00         | 0.01      | 0.00        |
> > > > > > | swiss42        | 42           | 0.00     | 0.00         | 0.00      | 0.00        |
> > > > > > | ts225          | 225          | 0.12     | 0.12         | 0.21      | 0.15        |
> > > > > > | tsp225         | 225          | 0.10     | 0.12         | 0.18      | 0.12        |
> > > > > > | u159           | 159          | 0.04     | 0.05         | 0.10      | 0.05        |
> > > > > > | u574           | 574          | 1.02     | 1.25         | 2.02      | 1.23        |
> > > > > > | u724           | 724          | 0.93     | 1.19         | 1.76      | 1.23        |
> > > > > > | u1060          | 1060         | 1.22     | 1.23         | 2.32      | 1.45        |
> > > > > > | u1432          | 1432         | 1.53     | 1.98         | 2.31      | 1.67        |
> > > > > > | u1817          | 1817         | 1.97     | 2.11         | 3.01      | 2.01        |
> > > > > > | u2152          | 2152         | 2.10     | 2.12         | 5.03      | 2.32        |
> > > > > > | u2319          | 2319         | 2.30     | 2.54         | 3.08      | 2.75        |
> > > > > > | ulysses16      | 16           | 0.00     | 0.00         | 0.00      | 0.00        |
> > > > > > | ulysses22      | 22           | 0.00     | 0.00         | 0.00      | 0.00        |
> > > > > > | vm1084         | 1084         | 1.32     | 1.70         | 2.47      | 1.46        |
> > > > > > | vm1748         | 1748         | 1.75     | 2.05         | 3.19      | 1.88        |

---

> > > > > > > ### Comment · Reviewer_su5g · 2023-08-18
> > > > > > >
> > > > > > > Thanks for your efforts. May I know what is the training problem size of each method?

---

> > > > > > > > ### Author Response · Authors · 2023-08-18
> > > > > > > >
> > > > > > > > Given a TSP instance of size $n$:
> > > > > > > >
> > > > > > > > For UTSP, we create 3000 unlabelled TSP instances with a size of $n$ for training. This process is fast as it doesn't require any label; we just generate 3000 randomly distributed sets of $n$ coordinates.
> > > > > > > >
> > > > > > > > In DIMES, the training dataset is generated on the fly. Throughout our training, we produce a total of 64000 TSP $n$ instances. This also fast by the fact that we don't necessitate the use of ground truth solutions for training.
> > > > > > > >
> > > > > > > > In Att-GCRN, as a supervised approach, the solver must be executed prior to training. We follow the method outlined by [Fu et al]. Their model is trained on **1 million** solved TSP-50 instances, and they utilize graph sampling to ensure adaptability across various sizes.
> > > > > > > >
> > > > > > > > In DIFUSCO, which is also a supervised method, we generate 50,000 solutions for each TSP instance sized $n$. Here, we use the LKH heuristic instead of an exact solver because an exact solver can be excessively time-consuming.
> > > > > > > >
> > > > > > > >
> > > > > > > > **Reference**:
> > > > > > > > Zhang-Hua Fu, Kai-Bin Qiu, and Hongyuan Zha. Generalize a small pre-trained model to arbitrarily large tsp instances.

---

### Official Review · Reviewer_wEAA · 2023-06-22

**Soundness:** 4 excellent
**Presentation:** 2 fair
**Contribution:** 4 excellent
**Rating:** 8
**Confidence:** 3

**Summary:**

The authors propose a continuous relaxation of the 2D Euclidian Travelling Salesman Problem. Specifically, they replace the combinatorial optimization problem with a continuous optimization problem of the form

*Minimize f(A) over all column-stochastic matrices $A \in \mathbb R^{n \times n}$*

where $f$ is a smooth function and $A$ is a soft indicator matrix whose $t^{th}$ column indicate the city visited at step $t$ of the tour. The matrix $A$ is obtained via a deep net $A = \phi_\theta(X)$ where $X$ contains the position of the cities, therefore leading to a problem of the form

*Minimize $f(\phi_\theta(X))$ over $\theta$*

which is optimized via SGD. From the soft indicator matrix $A$, the authors build a hard indicator matrix by performing a local search algorithm.

The proposed approach significantly differs from RL approaches, since it is based on optimizing a smooth objective. It is also significantly different from imitation learning approaches, in which a neural network is trained to imitate  the output of an accurate but slow solver (typically the Concorde algorithm, which is more or less the default TSP solver in the OR community).

The proposed approach outperforms previous RL approaches and imitation learning approaches.













**Strengths:**

* The proposed approach is simple and outperforms previous deep learning methods when applied to large TSP instance.
* The accuracy and sample efficiency achieved by the proposed approach are *very* impressive.
* The TSP problem is an important and prototypical combinatorial optimization problem. The proposed approach, due to its simplicity and performance, has the potential of becoming quite influential.



**Weaknesses:**

1.  A more extensive discussion of the existing literature is needed. In particular, the authors should explain the method presented in [1], as it is the only one that achieves results comparable to the proposed approach.  I would recommend to have a `related work' subsection.

2. The presentation could be improved. In particular I find that referring to the matrix $T$ as a transition matrix is very confusing. $H$ is a transition matrix, since $H_{ij}$ is to some extent the probability of moving from i to j. If I understand correctly the proposed relaxation, the $t^{th}$ column of $T$ is a probability distribution over the cities that indicates what is the city most likely to be visited at step $t$ of the tour. So I would refer to $T$ as a soft indicator matrix. The left side of Figure 3 is confusing: I take it as meaning that if $T$ is used as a transition matrix, then it leads to a non-Hamiltonian cycle. I think it is better to not encourage the reader to think of $T$ as a transition matrix. If $T=[p_1, \ldots, p_n]$ is viewed as a soft indicator matrix that indicates the city visited at time $t$, then the formula for $H$ can be written as a sum of outer products\
$
\displaystyle H = \sum_{t=1}^{n-1} p_{t} p_{t+1}^T + p_{n} p_{1}^T
$\
from which it is clear that $H$ is a transition matrix. I think equation (1) and the displayed matrix $V$ are unnecessary and take lots space. Similarly, figure 1, 2 and 3 take lots of space and do not bring much clarity in my opinion. I think the space could be better used by describing related works in more depth, adding context for the experimental results (which are very impressive), and also maybe discussing continuous relaxations in general.  Overall, the proposed strategy is very simple (which I see as a strength): I think it could be presented in a  more concise and cleaner manner.

3. The experimental section could be polished. I think it is important to explain what is the "gap" and the "rounding problem". (Also line 177 is hard to read, and the first paragraph is not well written.)





[1] Zhang-Hua Fu, Kai-Bin Qiu, and Hongyuan Zha. Generalize a small pre-trained model to arbitrarily
325 large tsp instances.

**Questions:**

Question1: On line 198 the authors write
> We remark that the UTSP takes a shorter total running time (inference + search) and outperform the existing learning baselines on these large instance.

 I am confused: the algorithm from [1] seems faster. Is it not a learning baseline? Again, I think more discussion about [1] is needed.


Question 2:  The GCN from Kipf and Welling is the `smoothest' GNN. Did you try other GNN/transformer architecture beyond SAG?

[1] Zhang-Hua Fu, Kai-Bin Qiu, and Hongyuan Zha. Generalize a small pre-trained model to arbitrarily
325 large tsp instances.


**Limitations:**

I did not find a paragraph about the limitation of the proposed approach. Overall this is a very strong work in my opinion. An honest explanation of the shortcomings of the proposed approach would improve the scientific contribution.

---

> ### Author Rebuttal · Authors · 2023-08-09
>
> *We update the experiment section in the one page rebuttal pdf.*
>
> **Q**: A more extensive discussion of the existing literature is needed. In particular, the authors should explain the method presented in [1], as it is the only one that achieves results comparable to the proposed approach. I would recommend to have a `related work' subsection.
>
> A: We will revise the introduction part and add a new section, refer to the "replies to all reviewers".
>
> **Q**: The presentation could be improved. In particular I find that referring to the matrix... I take it as meaning that if $\mathbb{T}$ is used as a transition matrix, then it leads to a non-Hamiltonian cycle.
>
> A:  **I take it as meaning that if $\mathbb{T}$  is used as a transition matrix, then it leads to a non-Hamiltonian cycle.** This is correct!
>
> We will change the transition matrix to  soft indicator matrix, we rewrite $\mathcal{H}$ as:
>
> $
> \mathcal{H} = \sum_{t=1}^{n-1}p_t p^T_{t+1} + p_np_1^T,
> $
>
> where $p_t \in \mathbb{R}^{n \times 1}$ is the $t_{th}$ column of $\mathbb{T}$, $\mathbb{T} = [p_1|p_2|...|p_n]$.}
>
> We remove $\mathbb{V}$, we will remove some figures (to the appendix)  and add more discussion on related works.
>
> **Q**: The experimental section could be polished. I think it is important to explain what is the "gap" and the "rounding problem".
>
> A: The gap: given a length and the optimal length $l_{opt}$, the gap is defined as:
> $(l-l_{opt})/l_{opt}$.
>
> Rounding problem: On many instances, the best known solutions reported by Concorde are not strictly optimal (confirmed in (Joshi et al., 2019), possibly due to round-off reasons), which could be slightly improved (< 10−2 ) by our algorithm (Fu et al., 2020).
>
> We will revise the experimental section and add above discussions.
>
> **Q**:  line 177 is hard to read:
>
> A: When we apply the T → H transformation and utilize H as the heat map, we observe that H forms a Hamiltonian Cycle. By minimizing the expression $\sum_{i=1}^{n} \sum_{j=1}^{n} D_{i,j} \cdot H_{i,j}$, we can achieve the shortest Hamiltonian Cycle, which represents the solution to the Traveling Salesman Problem (TSP).
>
> **Q**:  first paragraph is not well written:
>
> A: As mentioned above, we will revise the introduction part and add a related works section.
>
>
> Joshi et al., 2019: Chaitanya K Joshi, Thomas Laurent, and Xavier Bresson. An efficient graph convolutional network technique for the travelling salesman problem. arXiv preprint arXiv:1906.01227, 2019.
>
> Fu et al., 2020: Targeted sampling of enlarged neighborhood via Monte Carlo tree search for TSP
>
>
> **Q**: On line 198 the authors write
> We remark that the UTSP takes a shorter ...on these large instance...the algorithm from [1] seems faster. Is it not a learning baseline? Again, I think more discussion about [1] is needed.
>
> A: We will add more discussion on [1], Here we mean “large instances”like TSP  500 and 1000. As the problem size increases increases, our method performs better and take have a shorter running time, as summarized here (there is a typo is the TSP 500 column in the original manuscript, which overlap with the TSP 200, we update the performance in the one page pdf rebuttal):
>
> Results of SAG + Local Search w.r.t. existing baselines. We evaluate the models on 128 TSP 500 instances and 128 TSP 1000 instances.
>
> | Method | Type | TSP500 (Length) | TSP500 (Gap %) | TSP500 (Time) | TSP1000 (Length) | TSP1000 (Gap %) | TSP1000 (Time) |
> |--------|------|-----------------|----------------|---------------|------------------|-----------------|---------------|
> | Concorde | Solver | 16.5458 | 0.0000 | 37.66m | 23.1182 | 0.0000 | 6.65h |
> | Gurobi | Solver | 16.5171 | -0.1733 | 45.63h | - | - | - |
> | LKH3 | Heuristic | 16.5463 | 0.0029 | 11.41m | 23.1190 | 0.0036 | 38.09m |
> | GAT \cite{deudon2018learning} | RL, S | 28.6291 | 73.0293 | 20.18m | 50.3018 | 117.5860 | 37.07m |
> | GAT \cite{deudon2018learning} | RL, S 2-OPT | 23.7546 | 43.5687 | 57.76m | 47.7291 | 106.4575 | 5.39h |
> | GAT \cite{kool2018attention} | RL, S | 22.6409 | 36.8382 | 15.64m | 42.8036 | 85.1519 | 63.97m |
> | GAT \cite{kool2018attention} | RL, G | 20.0188 | 20.9902 | 1.51m | 31.1526 | 34.7539 | 3.18m |
> | GAT \cite{kool2018attention} | RL, BS | 19.5283 | 18.0257 | 21.99m | 29.9048 | 29.2359 | 1.64h |
> | GCN \cite{joshi2019efficient} | SL, G | 29.7173 | 79.6063 | 6.67m | 48.6151 | 110.2900 | 28.52m |
> | GCN \cite{joshi2019efficient} | SL, BS | 30.3702 | 83.5523 | 38.02m | 51.2593 | 121.7278 | 51.67m |
> | GCN \cite{joshi2019efficient} | SL, BS* | 30.4258 | 83.8883 | 30.62m | 51.0992 | 121.0357 | 3.23h |
> | DIMES \cite{qiu2022dimes} | RL+S | 18.84 | 13.84 | 1.06m | 26.36 | 14.01 | 2.38m |
> | DIMES \cite{qiu2022dimes} | RL+MCTS | 16.87 | 1.93 | 2.92m | 23.73 | 2.64 | 6.87m |
> | DIMES \cite{qiu2022dimes} | RL+AS+MCTS | 16.84 | 1.76 | 2.15h | 23.69 | 2.46 | 4.62h |
> | DIFUSCO \cite{sun2023difusco} |SL+MCTS | 16.63 | 0.46 | 10.13m | 23.39 | 1.17 | 24.47m |
> | Att-GCRN\cite{fu2021generalize} | SL+RL MCTS | 16.7471 | 1.2169 | 31.17s + 3.33m | 23.5153 | 1.7179 | 43.94s + 6.68m |
> | UTSP (Ours) | UL, Search | 16.6846 | 0.8394 | 1.37m + 1.33m | 23.3903 | 1.1770 | 3.35m + 2.67m |
>
> We now add DIMES and DIFUSCO, on TSP 500 and 1000, the new baseline DIMES is able to take a shorter running time, but results in a larger gap, we will revise the this section.
>
>
> **Q**: The GCN from Kipf and Welling is the `smoothest' GNN. Did you try other GNN/transformer architecture beyond SAG?
>
> A: We tried a graph attention network, but we didn't observe a significant improvement compared to GCN. This could be due to the fact that graph attention networks still rely on averaging features of neighboring nodes for similarity calculations.
> Developing specific GNNs targeted for solving TSP is a very important topic,  and we believe that there exists more suitable GNN structures, we will leave it for future discussions.
>
> **Q** limitation of the proposed approach.
>
> A: we will add more discussion about limitation and future works.

---

> > ### Comment · Reviewer_wEAA · 2023-08-11
> >
> > Thanks for the detailed answer. The added baseline DIFUSCO performs slightly better. Could you elaborate on this method (there is a single sentence in the added related work section). In particular, does it require TSP solutions computed by another algorithm such as Concorde? (i.e. is it trained on 'exact' TSP solutions)

---

> > > ### Author Response · Authors · 2023-08-11
> > > **Does DIFUSCO require TSP solutions computed by another algorithm such as Concorde?**
> > >
> > > Thank you for your comment,
> > >
> > > Yes, DIFUSCO necessitates TSP solutions derived from an alternative algorithm like Concorde.
> > >
> > >
> > > The authors mentioned that they use Concorde to generate the TSP  solutions, see Section 4.1 in https://arxiv.org/pdf/2302.08224.pdf.
> > >
> > > Please refer to the one page rebuttal .pdf file in the attachment,
> > > As shown in the "Type" column,   DIFUSCO is classified a supervised learning (SL) approach.
> > > More elaboration on DIFUSCO will be added in the related works section.

---

> > ### Comment · Reviewer_wEAA · 2023-08-13
> >
> > To me there is an important difference between algorithms that 'imitate Concorde' (that you refer to as SL) and algorithms that solve the TSP 'from scratch' (RL and unsupervised). The latter are more interesting because they are easier to adapt to new tasks. The former can only solve tasks for which a good combinatorial algorithm already exists; moreover, at best, they can only speed up the teacher algorithm. I think this point could be made stronger in the introduction.

---

> > > ### Author Response · Authors · 2023-08-14
> > >
> > > Thank you for your feedback. We will incorporate your suggestions into the manuscript and emphasize the significant difference between our unsupervised learning approach and other methods that rely on supervised/imitation learning.
> > >
> > > Specifically, we will underline the fact that approaches like DIFUSCO and Fu et al. require a dataset of over **1 million**  exactly solved TSP solutions for training on TSP 100.
> > >
> > > Strengthening this point in the introduction will further clarify the novelty and the contribution of our unsupervised method.

---

### Official Review · Reviewer_X3D7 · 2023-07-06

**Soundness:** 3 good
**Presentation:** 3 good
**Contribution:** 3 good
**Rating:** 6
**Confidence:** 4

**Summary:**

This paper proposed a novel unsupervised learning method for solving the Travelling Salesman Problem (TSP). It employs a Scattering Attention GNN (SAG) to encode the node information. Then, the learned representation is transformed into a heatmap, which corresponds to the probability of an edge being included in the optimal solution. A novel unsupervised loss is proposed. Based on the learned heatmap, a local search procedure with randomness is used to derive the final solution. Experiments on TSP instances up to 1000 nodes show that the proposed method can use much less training data, and outperforms several existing deep TSP models.

**Strengths:**

1. The research motivation is clear.

2. The idea of using SAG to build heatmap, and the proposed unsupervised loss is interesting.

3. Strong empirical performance, with detailed analysis on the expressive power of SAG.

4. Generally good writing and organization.

**Weaknesses:**

1. Since the proposed method is heatmap based, I suggest to give a detailed review and discussion on heatmap based methods, such as [Fu2021], [Qiu2022], [Joshi2022]. Currently the introduction is mainly from the SL/RL perspective.

2. For fair comprison, I suggest to add some ablation studies on the effect of local search. In the main results (Tables 1-3), it is unclear whether the performance improvement is from the better learning mechanism, or the local search procedure. In addition, the baselines use sampling, beam search and Monte carlo tree search as the decoding strategies, while the proposed method uses local search. This also makes it difficult to compare the effectiveness of the learning mechanism. While Figure 4, 5 and 6 provide some insights, they are still indirect results.

3. The baselines are not up-to-date. It would be much stronger if state-of-the-art methods, such as DIMES [Qiu2022] (also a heatmap based method), could be compared.

4. It is unclear how the training data is generated and whether the baselines are retrained. I guess they are not since GAT [Kool2019] and GCN [Joshi2019] is known to be hard to scale to problem of size 1000.

5. More detailed results on the training efficiency (e.g., a table) could be helpful. Currently only an informal discussion is given on Page 6.

6. The local search mechanism relies on randomness. But how it impacts the results is unclear, including how stable the performance is (std is helpful if reported), and what if no randomness is used.

7. The runtime in Table 2 seems incorrect. For the baselines, most runtime for TSP200 is much faster than TSP100. For your approach, the runtime for TSP200 is faster than TSP50. Please thoroughly check the results.

8. The proposed approach is currently limited to TSP. It would be better to have some discussions on how to extend it to other typical routing problems such as CVRP.


**Questions:**

Please see the above weaknesses.

**Limitations:**

The propossed approach is limited to TSP. But this is not a severe limitation since it obtains good results on large-scale problems. It would be better to have some discussions on how to generalize the idea to support other routing problems.

---

> ### Author Rebuttal · Authors · 2023-08-10
>
> Please check the “replies to all reviewer” and the one page rebuttal pdf.
>
> **Q** Since the proposed method is heatmap based, I suggest to give a detailed review and discussion on heatmap based methods, such as [Fu2021], [Qiu2022], [Joshi2022]. Currently the introduction is mainly from the SL/RL perspective.
>
> A: We will  add a related works section and will revise the introduction.
>
> **Q** For fair comprison, I suggest to add some ablation studies on the effect of local search. In the main results (Tables 1-3), it is unclear whether the performance improvement is from the better learning mechanism, or the local search procedure. In addition, the baselines use sampling, beam search and Monte carlo tree search as the decoding strategies, while the proposed method uses local search. This also makes it difficult to compare the effectiveness of the learning mechanism. While Figure 4, 5 and 6 provide some insights, they are still indirect results.
>
> A: We include ablation studies to examine how changes in the search hyperparameters affect performance in the one page rebuttal pdf. This is shown in "Table: Search Hyperparameter Ablation Study on TSP 100" and "Table: Search Hyperparameter Ablation Study on TSP 1000". A lower value of $\alpha$ indicates that the local search algorithm prioritizes edges with higher heat map values, whereas a higher value of $\alpha$ aligns more with an MCTS style, which is similar to the approach described in [Fu et al., 2021].
>
> We also find a better performance on TSP 100, where the gap is -0.0019 %.
>
> **Q**  The baselines are not up-to-date. It would be much stronger if state-of-the-art methods, such as DIMES [Qiu2022] (also a heatmap based method), could be compared.
>
> A: we add new baselines, refer to the one page rebuttal pdf.
>
> **Q** It is unclear how the training data is generated and whether the baselines are retrained. I guess they are not since GAT [Kool2019] and GCN [Joshi2019] is known to be hard to scale to problem of size 1000.
>
> A: Regarding the training data, our approach aligns with [Fu et al.]. The training data is randomly generated on a 2D plane, and we adopt the same test data set as outlined in their work.
> For TSP instances with sizes 20, 50, and 100, the test data comprises 10,000 automatically generated 2D Euclidean TSP instances for each size. This test data set is widely utilized by existing learning-based algorithms.
> On larger instances with n = 200, 500, and 1000, There are 128 instances for each size.
>
>
>
>
> Regarding retrain:
>
> See our manuscript, Line 395 to line 398: For UTSP (our method) and the state-of-the-art learning-based method Att-GCRN Fu et al. [2021], we run the search algorithm on **exactly the same environment (one Intel Xeon Gold 6326)** for a fair comparison.
>
> And for other baselines, since GAT [Kool2019] and GCN [Joshi2019] have a noticeable gap from Fu et al. [2021], we directly refer to the results from Fu et al. [2021].
>
>
>
>
> **Q**:  More detailed results on the training efficiency (e.g., a table) could be helpful. Currently only an informal discussion is given on Page 6.
>
> A: We will add more discussion in the new manuscript.
>
> **Q**: The local search mechanism relies on randomness. But how it impacts the results is unclear, including how stable the performance is (std is helpful if reported), and what if no randomness is used.
>
> A: Here we compare the performance using restart and without restart on  TSP 100 and TSP 1000, all other hyperparameters are the same.
>
> On TSP 100, using restart, the gap is -0.0019% and the std is 0.00035%.
>
> On TSP 100, without restart, the gap is -0.0017% and the std is 0.00025%.
>
> On TSP 1000, using restart, the gap is 1.1770% and the std is 0.075
>
> On TSP 1000, without restart, the gap is 1.2163% and the std is 0.38.
>
> The results indicate that introducing randomness (via restarts) helps enhance performance, and has the potential to eliminate the standard deviation in larger TSP instances, such as TSP 1000.
>
>
> **Q**: The runtime in Table 2 seems incorrect. For the baselines, most runtime for TSP200 is much faster than TSP100. For your approach, the runtime for TSP200 is faster than TSP50. Please thoroughly check the results.
>
> A: We evaluate our performance on the same test dataset used in [Fu et al.] and [Qiu et al.].
> In that dataset, TSP instances with sizes 20, 50, and 100 have **10,000** test samples each.
> However, for TSP instances with sizes 200, 500, and 1000, we have a smaller set of only **128** samples.
>
> That’s why TSP200 is faster than TSP50 and TSP 100.
>
> The new performance is included in the table one page rebuttal pdf.
>
>
> **Q** The proposed approach is currently limited to TSP. It would be better to have some discussions on how to extend it to other typical routing problems such as CVRP.
>
> A: A common aspect is that both CVRP and TSP share the objective of minimizing the 'distance'. As a result, we can employ expressions such as $\sum_{ij} H_{ij}D_{ij}$. We will add a future work section in our manuscript.
>
>
>
> Reference:
>
> Fu et al. Generalize a small pre-trained model to arbitrarily large tsp instances.
> Proceedings of the AAAI Conference on Artificial Intelligence, 35(8):7474–7482, 2021
>
> Qiu et al. Dimes:
> Dimes: A differentiable meta solver for combinatorial optimization problems." Advances in Neural Information Processing Systems 35 (2022): 25531-25546.

---

> > ### Comment · Reviewer_X3D7 · 2023-08-14
> >
> > Thanks for the response. While it addressed most of my concerns, I am not quite satisfied with the ablation results on local search. What I meant is that, in Table 1-3 in the main paper and Table 1-2 in the rebuttal PDF, different methods are combinations of different learning mechanisms + search mechanisms. Taking DIMES as an example, it is RL + MCTS, while the proposed UTSP is UL + Search. Since the search mechanisms are different (MCTS vs local search), it is unclear whether the improvement over DIMES comes from the newly proposed learning mechanism, or the local search procedure.
> >
> > Besides, I strongly recommend to have a detailed comparison on the training time with the SL/RL baselines (including label collection time for SL), so as to better justify the main motivation of this paper, i.e. the proposed UL method significantly reduces training cost and label requirements.

---

> > > ### Author Response · Authors · 2023-08-17
> > > **MCTS vs local search**
> > >
> > > Thank you for your clarification.
> > >
> > > Here we use change our search to the MCTS and this is the result. We use the same time budget for running both MCTS and the search method.
> > >
> > > In general, our search method outperform MCTS, albeit not by a substantial margin. And we can see that when using MCTS, our UL method still outperforms DIMES, so the improvement over DIMES comes from the newly proposed learning mechanism.
> > >
> > > | Method                                               | Type            | TSP20 Length | TSP20 Gap (%) | TSP20 Time | TSP50 Length | TSP50 Gap (%) | TSP50 Time | TSP100 Length | TSP100 Gap (%) | TSP100 Time |
> > > | ---------------------------------------------------- | --------------- | ------------ | ------------- | ---------- | ------------ | ------------- | ---------- | ------------- | -------------- | ----------- |
> > > | Concorde                                             | Solver          | 3.8303       | 0.0000        | 2.31m      | 5.6906       | 0.0000        | 13.68m     | 7.7609        | 0.0000         | 1.04h       |
> > > | LKH3                                                 | Heuristic       | 3.8303       | 0.0000        | 20.96m     | 5.6906       | 0.0008        | 26.65m     | 7.7611        | 0.0026         | 49.96m      |
> > > | Att-GCRN                    | SL+RL+MCTS      | 3.8300       | -0.0074       | 1.44m  | 5.6908       | 0.0032        | 5.22m     | 7.7616        | 0.0096         | 9.19m     |
> > > | DIMES                           | RL+AS+MCTS      | 3.8304       | 0.0026        | 2.4h       | 5.6919       | 0.0232        | 10.3h      | 7.7654        | 0.05772        | 32.5h       |
> > > | DIFUSCO                 | SL+MCTS         | 3.8303       | 0.0012        | 3.34m      | 5.6908       | 0.0029        | 10.13m     | 7.7612        | 0.00386        | 19.15m      |
> > > | UTSP (ours) - MCTS                                  | UL, MCTS        | 3.8303       | -0.0012      | 1.67m  | 5.6899       | -0.0123       | 3.94m    | 7.7608        | -0.0007        | 10.89m     |
> > > | UTSP (ours) - Search                                | UL, Search      | 3.8303       | -0.0009       | 1.67m    | 5.6894       | -0.0200       | 3.94m    | 7.7608        | -0.0019        | 10.89m      |
> > >
> > >
> > >
> > >
> > >
> > >
> > >
> > >
> > > | Method                            | Type           | TSP200 Length | TSP200 Gap (%) | TSP200 Time | TSP500 Length | TSP500 Gap (%) | TSP500 Time | TSP1000 Length | TSP1000 Gap (%) | TSP1000 Time |
> > > | --------------------------------- | -------------- | ------------- | -------------- | ----------- | ------------- | -------------- | ----------- | -------------- | --------------- | ------------ |
> > > | Concorde                          | Solver         | 10.7191       | 0.0000         | 3.44m       | 16.5458       | 0.0000         | 37.66m      | 23.1182        | 0.0000          | 6.65h        |
> > > | LKH3                              | Heuristic      | 10.7195       | 0.0040         | 2.01m       | 16.5463       | 0.0029         | 11.41m      | 23.1190        | 0.0036          | 38.09m       |
> > > | Att-GCRN  | SL+RL+MCTS     | 10.7358       | 0.1563         | 1.67m    | 16.7471       | 1.2169         | 3.85m    | 23.5153        | 1.7179          | 7.41m    |
> > > | DIMES        | RL+AS+MCTS     | 10.7403       | 0.1977         | 46m         | 16.84         | 1.76           | 2.15h       | 23.69          | 2.46            | 4.62h        |
> > > | DIFUSCO    | SL+MCTS        | 10.7521       | 0.3079         | 4.12m       | 16.63         | 0.46           | 10.13m      | 23.39          | 1.17            | 24.47m       |
> > > | UTSP (Ours)                | UL, MCTS       | 10.7312       | 0.1129         | 1.67m      | 16.7026       | 0.9477         | 2.70m      | 23.4729        | 1.5343          | 6.02m   |
> > > | UTSP (Ours)              | UL, Search     | 10.7289       | 0.0918         | 1.67m    | 16.6846       | 0.8394         | 2.70m      | 23.3903        | 1.1770          | 6.02m     |

---

> > > ### Author Response · Authors · 2023-08-17
> > > **Total running time comparison**
> > >
> > > Thank you for your clarification.
> > >
> > > Here, we are comparing the total time cost. Since Att-GCRN and DIFUSCO are supervised methods, generating the ground truth label is already a very time-consuming process, especially as the size grows larger. DIMES uses reinforcement learning (RL) and does not require ground truth labels, resulting in a smaller total time cost.
> > >
> > > Our UL method has the smallest time cost.
> > >
> > >
> > >
> > >
> > > | TSP SIZE | Att-GCRN (supervised)      | DIFUSCO (supervised)    | DIMES (reinforcement) | UTSP (unsupervised) |
> > > | -------- | -------------------------- | ----------------------- | --------------------- | ------------------- |
> > > | 20       | 3.8h (solver) + 12h        | 5.8h (solver) + 5h     | 2.9h                  | 27m                 |
> > > | 50       | 22.8h (solver) + 25h       | 34.3h (solver) + 6h    | 11.1h                 | 36m                 |
> > > | 100      | 22.8h (solver) + 25h       | 156.2h (solver) + 8h   | 33.5h                 | 55m                 |
> > > | 200      | 22.8h (solver) + 25h       | 33.5h (solver) + 9h    | 2.5h                  | 1.2h                |
> > > | 500      | 22.8h (solver) + 25h       | 190h (solver) + 13h    | 3.6h                  | 1.8h                |
> > > | 1000     | 22.8h (solver) + 25h       | 317h (solver) + 16h    | 6.3h                  | 2.1h                |
> > >
> > >
> > >
> > >
> > > It's important to mention that for Att-GCRN, their training was performed on TSP 20 and 50, and they employed graph sampling to construct small heat maps and then merge them. Consequently, the time cost of TSP 100, 200, 500, and 1000 are the same.
> > >
> > > For DIFUSCO, the original paper train the model with 8× NVIDIA Tesla V100 Volta GPUs, here we train on single one V100. However, since DIFUSCO is a supervised learning method, running the solver is already very time consuming.

---

> > > > ### Comment · Reviewer_X3D7 · 2023-08-17
> > > >
> > > > Thanks for the new results, which show that the proposed method indeed significantly saves the training and label collection cost, while delivering competitive solution quality. I increased my score to 6.

---

> > > > > ### Author Response · Authors · 2023-08-18
> > > > >
> > > > > Thank you for the insightful discussion. We greatly appreciate your suggestions and will incorporate them into the manuscript.

---

### Official Review · Reviewer_vHRb · 2023-07-11

**Soundness:** 2 fair
**Presentation:** 3 good
**Contribution:** 1 poor
**Rating:** 4
**Confidence:** 5

**Summary:**

The paper proposes an unsupervised learning-based heuristic to solve the Travelling Salesman Problem. The approach consists of two steps: first a GNN is trained using a surrogate loss to output a heatmap of the edges then the heatmap is used to guide a local search heuristic. The proposed model has significantly less parameters than similar previous methods and its training is very efficient. It is tested on TSP instances with up to 1000 nodes.

**Strengths:**

1. The paper introduces a light-weight model and sample-efficient training procedure to learn a TSP heuristic
1. Novel idea of a differentiable surrogate loss that encourages the solution to form a hamiltonian cycle
1. Nice discussion and motivation of the choice of the GNN and precise comparison between GCN and SAG
1. The paper is clear and easy to follow

**Weaknesses:**

1. The main weakness is the limited scope: the paper is very specialized to the TSP, esp. the proposed surrogate loss is specific to the TSP.
1. Missing some related work: e.g. [1] and [2] and the associated baselines in the experiments. These are known to be much stronger baselines than the ones presented in Tables 1-3.
1. Some unfounded claims:
    * The paper says “Such SL models scale poorly to the big instances” while [Fu et al 2021] scales remarkably well to instances with up to 10,000.
    * L198 “We remark that the UTSP takes a shorter total running time (inference + search) and outperform the existing learning baselines on these large instances. ” But the reported computation times are not shorter than Att-GCRN for TSP200 and TSP50.
    * L216: “when using SL, the model learns from the TSP solutions, which fails when multiple solutions exist or the solutions are not optimal.” Is there any proof/reference to motivate this statement?
    * L194 “On larger instances with n = 200, 500 and 1, 000, we notice that traditional solvers and heuristics (Concorde,Gurobi and LKH3) fail to generate the optimal solutions within reasonable time when the size of problems grows”. I’m not sure what is meant exactly, since we see for e.g. in Table 3 that LKH still gives the best optimality gap for TSP1000 in 38minutes. Then for Gurobi, I wonder what was the time limit. Since it is an exact solver, it is expected to be slow. However it should return the best solution it found when it reaches the time limit. Can the authors give more details about the results (or lack of) with Gurobi.
1. The columns TSP200 in Tab 2 and TSP500 in Tab 3 look exactly the same. Is it a typo?
1. Many components in the approach and there is only one kind of ablation for the choice of SAG versus GCN for the model architecture. It would be interesting to do precise ablations, esp. for the quality and role of the heatmap in the final results. For example by applying the same local search strategy with the heatmap obtained by other models such as the GCN approach of [Joshi et al 2019a] or the Att-GCRN of [Fu et al 2021].
1. Experiments only on synthetic instances of the same distribution as training. Could be interesting to test on unseen distributions, e.g. TSPlib.


[1] Qiu et al, DIMES: A Differentiable Meta Solver for Combinatorial Optimization Problems, NeurIPS 2022

[2] Kwon et al POMO: Policy Optimization with Multiple Optima for Reinforcement Learning, NeurIPS 2020

**Questions:**

1. The paper says “Such SL models scale poorly to the big instances” while the approach of [Fu et al 2021] scales remarkably well to instances with up to 10,000. Maybe the authors meant *training* such SL models does not scale?

1. Have the authors tried their model on even larger instances, such as TSP10000?

1. Would the approach apply to the non-Euclidian asymmetric version of the TSP?

1. It would be useful to discuss how the temperature parameter (L73) and the number M (of elements to keep in each row of the heatmap) are fixed. In particular, how is the search space reduction (Sec. 5) affected by the choice of M?

1. L156 “when selecting the city v given u , we only consider the cities from the candidate set of v” —> do you mean the candidate set of *u* instead?

1. “the negative values are the results of the rounding problem” can the authors elaborate on what is meant by the rounding problem?

**Limitations:**

The limitations are not explicitly mentioned in the paper.

---

> ### Author Rebuttal · Authors · 2023-08-09
>
> **Q**: The main weakness is the limited scope: the paper is very specialized to the TSP
>
> A: TSP stands as one of the 21 NP-complete problems outlined by Karp [Karp].  TSP holds a foundational position in the field of combinatorial optimization owing to its  essence and practical utility. TSP is also fundamentally important within theoretical computer science and operations research domains. In this paper, we demonstrate that we can build unsupervised learning-based heuristics for the TSP, without the need for labeled ground truth solutions (supervised/imitation learning) or reliance on the framework of Reinforcement Learning (RL). We consider this to be a significant advancement.
>
> [Karp] Karp, Richard M. (1972). "Reducibility Among Combinatorial Problems"
>
>
> **Q**: Missing some related work:
>
> A: we add new baselines and section, refer to the table in the one page rebuttal pdf.
>
> **Q**: The paper says “Such SL models scale poorly to the big instances” while [Fu et al 2021] scales remarkably well to instances with up to 10,000.
>
> A: In [Fu et al.], the model is trained on the small graph and only generates a small heat map, such as for 20 cities or 50 cities. They train the model (in a supervised manner) at a small scale. They apply a series of techniques, such as graph sampling, graph conversion, and merging heat maps. This means that when dealing with large instances such as TSP 200, 500, and 1000, their model always generates a heat map of size 50 by 50, and then the sub heat maps are merged together. Please refer to the Methodologies section in [Fu et al.] for more details.
>
>
> **Q**: L198 “...not shorter than Att-GCRN for TSP200 and TSP50.
>
>
> A: “On large instances”we mean TSP 500 and TSP 1000, refer to the table in the one page rebuttal pdf., we will revise this sentence and add more discussion on the performance.
>
>
>
> **Q**: L216: “when using SL, ...Is there any proof/reference to motivate this statement?
>
> A: In [Li et al.], Section 4.2
> ... when there are multiple optimal solutions for the same graph. ..., two equivalent optimal solutions that induce completely different labellings., …which is not a useful labelling.
> We will add [Li et al.] into reference.
>
> [Li et al.] "Combinatorial optimization with graph convolutional networks and guided tree search." Advances in neural information processing systems 31 (2018). https://arxiv.org/pdf/1810.10659.pdf
>
> **Q**: L194 ...Can the authors give more details about the results (or lack of) with Gurobi.
>
> A: On TSP 1000 we tried set time limit as LKH running time (38.09m), Gurobi returns a ~18% Gap.
>
> **Q**: The columns TSP200 in Tab 2 and TSP500 in Tab 3 look exactly the same. Is it a typo?
>
> A: We fix the typo, we update the performance in the one page rebuttal pdf.
>
> **Q**: Many components in the approach and there is only one kind of ablation...
>
> A: We include ablation studies to examine how changes in the search hyperparameters affect performance. This is shown in "Table: Search Hyperparameter Ablation Study on TSP 100" and "Table: Search Hyperparameter Ablation Study on TSP 1000". A lower value of $\alpha$ indicates that the local search algorithm prioritizes edges with higher heat map values, whereas a higher value of $\alpha$ aligns more with an MCTS style, similar to the approach described in [Fu et al., 2021].
>
> **Q**: Experiments only on synthetic instances of the same distribution as training. Could be interesting to test on unseen distributions, e.g. TSPlib.
>
> A: The objective of this paper is to demonstrate the applicability of UL for the TSP. To assess our model's performance, we conducted evaluations on the same datasets used in [Fu et al., 2021], which serves as our primary baseline. We leave the exploration of TSPlib datasets to future research work.
>
> **Q**: Have the authors tried their model on even larger instances, such as TSP10000?
>
> A: Regarding TSP-10000, It is important to highlight that for the TSP-10000 evaluation, these studies [1, 2, 3] also use the same test dataset containing only 16 samples. Consequently, due to the limited size of the dataset, the performance results may not be a reliable indicator. Using the graph sampling technique suggested in [3], we evaluate UTSP and achieve ~3.05% gap in approximately 1 hour. For reference, [3] reported a 4.3% gap in 21 minutes, DIMES achieved a 4.0% gap in 30 minutes, and 3.2% gap in 3.5 hours. DIFUSCO outperforms all others, as it reports the best performance with a 2.5% gap and a time cost of 47 minutes.
>
> [1] DIMES: A differentiable meta solver for combinatorial optimization problems.
> [2] DIFUSCO: Graph-based diffusion solvers for combinatorial optimization.
> [3] Generalize a small pre-trained model to arbitrarily large tsp instances.
>
>
> **Q**: Would the approach apply to the non-Euclidian asymmetric version of the TSP?
>
> A: Modify the distance matrix $\mathcal{D}$ in Equation (2) to make it asymmetric.
>
>
> **Q**: It would be useful to discuss ... how is the search space reduction (Sec. 5) affected by the choice of M?
>
> A: We will add discussion on that, overall, increasing $M$ will have more overlapped cases. But then the search space is larger because we consider more possible edges.
>
> **Q**:  L156 “when selecting the city v given u ...
>
> A: Here the candidate set of v means We only select city v from the top M heat map value or the nearest M cities. We will revise this sentence.
>
>
> **Q**: “the negative values are the results of the rounding problem”... what is meant by the rounding problem?
>
> A: Rounding problem: On many instances, the best known solutions reported by Concorde are not strictly optimal (confirmed in (Joshi et al., 2019), possibly due to round-off reasons), which could be slightly improved (< 10−2 ) by our algorithm (Fu et al., 2020).
>
> Fu et al., 2020: Targeted sampling of enlarged neighborhood via Monte Carlo tree search for TSP

---

> > ### Comment · Area_Chair_z5Cf · 2023-08-19
> > **RE:**
> >
> > The author's rebuttal attempts to address your primary concerns. Do you have further questions or do your concerns still stand?

---

> > > ### Comment · Reviewer_vHRb · 2023-08-21
> > > **Thanks for the rebuttal**
> > >
> > > I thank the authors for their answers to my questions and the corrections of the results and added baselines.
> > > After reading the rebuttal and the other reviews, my main concern, which is the limited scope of the paper, still holds.
> > > I will increase my score to 4.

---

### Author Rebuttal · Authors · 2023-08-10

Dear Reviewers, thank you for your comments.

We updated our model's performance in the tables included in the one-page rebuttal PDF.

We have incorporated additional baselines: POMO by Kwon et al. [2020] and more recent approaches such as DIMES by Qiu et al. [2022] and DIFUSCO by Sun and Yang [2023].

There is a typ in the TSP 500 column of the original manuscript. We have now corrected this error and updated the correct value, refer to the attached one page pdf.

In one-page rebuttal PDF, we also include Search Hyperparameter ablation study,  an illustration of the Search Process and the overall structure of SAG. We also found better performance on TSP 100.

We will also update the manuscript, rewrite the introduction part and add a “related works” section to include recent advancements in data-driven methods for the Traveling Salesman Problem (TSP). Here is a preview of the proposed content for the "related works" section:

# Related Works

Researchers have been exploring the application of RL and SL techniques to tackle the TSP [Joshi 2022]. For example, [Kwon, 2020] uses a data-driven approach known as Policy Optimization with Multiple Optima (POMO), which relies on RL and avoids the utilization of hand-crafted heuristics. [Qiu, 2022] proposes a Meta-Learning framework that enhances the stability of REINFORCE-based training. [Sun 2023] applies SL and adopts a graph-based diffusion framework. Additionally, they introduce a cosine inference schedule to improve the efficiency of their model. [Fu 2021] also uses SL in their approach. They incorporate a heat map-based technique into an end-to-end model. Furthermore, they leverage graph sampling to extract small sub-graphs from the initial large graph. Subsequently, they train the GNN model on these sub-graphs to generate the corresponding heat maps, represented as probability matrices over the edges. Finally, the authors merge all the individual heat maps to create the final heat map.


Reference:

[Joshi 2022]:  Joshi, Chaitanya K., et al. "Learning the travelling salesperson problem requires rethinking generalization." Constraints 27.1-2 (2022): 70-98.

[Kwon, 2020]: Kwon et al. (2020). “Policy optimization with multiple optima for reinforcement learning.” Advances in Neural Information Processing Systems 33 (2020): 21188-21198.

[Qiu, 2022]: “A differentiable meta solver for combinatorial optimization problems.” Advances in Neural Information Processing Systems 35 (2022): 25531-25546.

[Sun 2023]: “Graph-based diffusion solvers for combinatorial optimization.” arXiv preprint arXiv:2302.08224 (2023).

[Fu 2021]: “Generalize a small pre-trained model to arbitrarily large tsp instances.” Proceedings of the AAAI conference on artificial intelligence. Vol. 35. No. 8. 2021.

---

### Decision · Program_Chairs · 2023-09-21

**Decision:**

Accept (poster)

**Comment:**

After the rebuttal, the remaining concerns were:
* Limit scope raised by reviewer vHRb and agreed on by the other reviewers.
* The experiments do not decouple the gains from the unsupervised approach to generating the heatmaps and the local search. Several reviewers asked for this and I and the reviewers did not find the authors answer satisfactory.

Two reviewers were borderline/leaning to accept, one voted for rejection and one championed the paper saying, "The proposed method is simple and state-of-the-art. [...] Even if it is very specific to the TSP, it shows that a straightforward continuous relaxation of a combinatorial problem, combined with a neural net, can lead to excellent results. I think it is likely to inspire other researchers."

Given that the primary concern is around narrow problem scope rather than a strong technical weakness and that several of the reviewers found the work to be of interest, I'm inclined to recommend acceptance.